# Geodesic Multi-Modal Mixup for Robust Fine-Tuning

**Changdae Oh**[*]
University of Seoul

**Junhyuk So**[*]
POSTECH

**Hoyoon Byun**
University of Seoul

**YongTaek Lim**
University of Seoul

**Minchul Shin**
KAIST

**Jong-June Jeon**
University of Seoul

**Kyungwoo Song**[+]
Yonsei University

## Abstract

Pre-trained multi-modal models, such as CLIP, provide transferable embeddings and show promising results in diverse applications. However, the analysis of learned multi-modal embeddings is relatively unexplored, and the embedding transferability can be improved. In this work, we observe that CLIP holds separated embedding subspaces for two different modalities, and then we investigate it through the lens of *uniformity-alignment* to measure the quality of learned representation. Both theoretically and empirically, we show that CLIP retains poor uniformity and alignment even after fine-tuning. Such a lack of alignment and uniformity might restrict the transferability and robustness of embeddings. To this end, we devise a new fine-tuning method for robust representation equipping better alignment and uniformity. First, we propose a *Geodesic Multi-Modal Mixup* that mixes the embeddings of image and text to generate hard negative samples on the hypersphere. Then, we fine-tune the model on hard negatives as well as original negatives and positives with contrastive loss. Based on the theoretical analysis about hardness guarantee and limiting behavior, we justify the use of our method. Extensive experiments on retrieval, calibration, few- or zero-shot classification (under distribution shift), embedding arithmetic, and image captioning further show that our method provides transferable representations, enabling robust model adaptation on diverse tasks. Code: https://github.com/changdaeoh/multimodal-mixup

## 1 Introduction

Witnessing the remarkable success of large-scale pre-training approaches, there have been numerous attempts to build a single *general-purpose model* (so-called foundation model [1]) rather than casting multiple task-specific models from scratch. Once built, it can be adapted (e.g., fine-tuned) to a wide variety of downstream tasks by leveraging its transferable representation. To construct such a general-purpose model, besides architectural design [2, 3, 4] and datasets [5, 6, 7], learning methods [8, 9, 10, 11, 12] have been known to be crucial for inducing transferable representation and for adapting the model robustly.

Contrastive Learning (CL) [13, 14, 15] is one of the most popular learning methods that constructs a discriminative embedding space by minimizing distances between positive pairs while maximizing them between negative pairs. Based on its versatility, CL has been widely adopted to (pre-) train models on various domains [15, 16, 17, 18]. Beyond CL on a single modality, CLIP [19] popularized

---

[*]Equal contribution (changdae.oh@uos.ac.kr; junhyukso@postech.ac.kr), [+]Corresponding author

37th Conference on Neural Information Processing Systems (NeurIPS 2023).

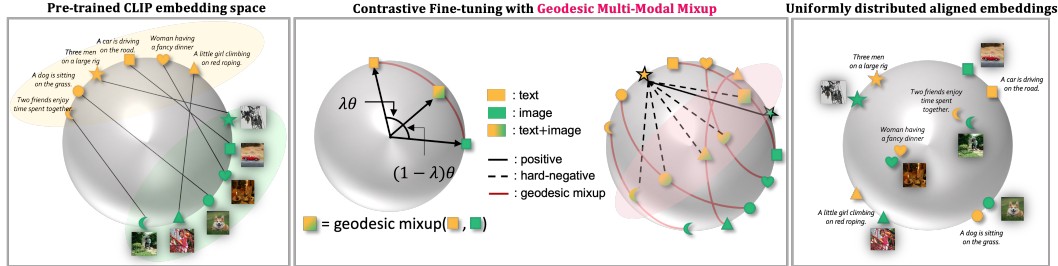

Figure 1: (Left) pre-trained CLIP has two separated clusters for image and text, with a large unexplored interspace. As a result, it has poor uniformity and alignment that might limit the embedding transferability. (Middle) For robust representation, our geodesic multi-modal Mixup ($m^2$-Mix) explores the unexploited interspace by mixing the heterogeneous embeddings on hypersphere. Generated samples by $m^2$-Mix are regarded as hard negatives for contrastive loss. (Right) As a result, fine-tuning with $m^2$-Mix induces robust representation with better transferability.

*multi-modal CL*, which aims to produce close embeddings for paired image-text instances and dissimilar ones for non-paired instances. Due to its generalization capability and transferability, pre-trained CLIP and its embeddings have been employed on various downstream tasks [20, 21, 22, 23].

However, we observed an unexpected phenomenon: while CLIP's learning objective is designed to align the image and text embeddings explicitly, it has two separate subspaces for each modality, and there are large unexplored interspaces between them as in Fig. 1 (illustration) and Fig. 2 (DOSNES [24] visualization). We further analyzed this through the lens of *uniformity-alignment* [25, 26], which is well-studied under uni-modal CL settings but unexplored on multi-modal CLs, and found that CLIP has a poor uniformity-alignment (Fig. 2 middle) due to its bipartite embedding structure. Theoretically and empirically, we confirmed that this property is unchanged even after fine-tuning. As discussed by Wang et al. [25], low uniformity-alignment may limit the embedding transferability. Liang et al. [27], concurrently (in terms of ArXiv preprints) took a similar observation, *modality gap*, which incurs a bunch of following works, and they also found that increasing temperature ($\tau$) in contrastive loss could somewhat reduce the embedding separation. However, varying the temperature requires manual engineering for each downstream task, and it incurs an inevitable trade-off between uniformity-alignment [26]. It raises an important question motivating this work: *"How can we obtain a multi-modal representation dealing better with the uniformity-alignment for robust transfer?"*

To answer this, we propose a fundamental learning method, **geodesic multi-modal Mixup** ($m^2$-Mix). As shown in Fig. 1, $m^2$-Mix blends the embeddings of different domains, e.g., image and text, and utilizes the mixtures as new entrees of contrastive loss. Our $m^2$-Mix contributes to robust representation learning in three perspectives. First, it generates hard negative samples which have high similarity with positives (incentivizing alignment), and they are known to be crucial for robust CL [28, 29, 26]. We provide a theoretical guarantee of the hardness of $m^2$-Mixed samples with empirical results. Second, $m^2$-Mix interpolates samples from heterogeneous domains to expand the effective embedding space (increasing uniformity) untapped by pre-trained CLIP, and this is further supported by the limiting behavior of $m^2$-Mix (see Prop. 4.2). Third, $m^2$-Mix produces virtual augmented samples in embedding space, so it endows a better perception on the out-of-distribution samples as well as in-distribution samples than standard CL.

Contributions: (1) We found that CLIP has a bipartite embedding space with poor uniformity-alignment, which may limit the embedding transferability for downstream tasks. (2) To address this, we propose a new fine-tuning method based on geodesic multi-modal Mixup, $m^2$-Mix. To our knowledge, this is the first work that adopts direct mixtures between heterogeneous embeddings. (3) We devised $m^2$-Mix as a geodesic Mixup to ensure the mixed embeddings are on a hypersphere, and this is beneficial for stable learning with $L_2$-normalized embeddings. (4) We validate our method by theoretical analyses and extensive experiments on retrieval, calibration, full-/few-shot classification (under distribution shift), embedding arithmetic, and image captioning.

## 2 Related Works

**Contrastive Representation Learning**  Contrastive Learning (CL) has been widely utilized for representation learning on various domains [15, 17, 30, 19, 31]. The goal of CL is to learn an embedding function that maps data into an embedding space so that semantically similar data

have close embeddings. Most of CLs adopt $L_2$-normalized embedding [13, 14, 15, 19] for stable learning [32], and it makes the embeddings lie on a unit hypersphere. There are key properties to analyze the hyperspherical embeddings, so-called *Uniformity and Alignment* [25]. Better uniformity and alignment can be regarded as a higher embedding transferability, so it is related to performance on downstream tasks. However, uniformity-alignment analysis on multi-modal settings has not been sufficiently explored yet [33], and we found that CLIP has poor uniformity-alignment before and even after fine-tuning. Meanwhile, it is known that hard negatives for constructing contrastive pairs are necessary to increase the robustness of contrastive representation learning [34, 35, 26, 36]. To this end, we devise a new approach for multi-modal CL, which generates hard negatives and achieves better uniformity-alignment for *robust and transferable* representation.

**Mixup**   There have been numerous papers that claim Mixup [37] is helpful for robust representation learning and alleviates the overconfident problems and failure under distribution shift as well as the in-distribution accuracy [38, 39, 40]. Based on such success of Mixup, many works are adopting it as a component of learning frameworks (on vision [41, 42, 43, 44], language [45, 46, 47] and graph [48, 49]). Besides, CL with Mixup to help representation learning has also been studied. While traditional CL annotates as 1 for positive pairs and 0 for negative pairs, $i$-Mix [50] and Un-Mix [51] interpolate the images with ratio $\lambda$, and adopt contrastive loss with pseudo labels according to the mixing ratio $\lambda$. However, there are few works on Mixup for multi-modal learning [52, 53]. STEMM [52] mixes the speech and text features to align them in shared embedding space, but it cannot be widely used for multi-modal learning because of its architecture-specific design. Therefore, we propose $m^2$-Mix, that can be broadly adopted for robust multi-modal representation learning.

**Multi-modal Learning**   To build a universal intelligence system that simultaneously processes multi-modal data streams, some focus on developing unified frameworks for multi-modal perception [2, 3, 54, 4, 55], others aim at representation learning under multi-modal correspondence [19, 31, 56, 57]. This work focuses on CLIP [19], the representative multi-modal representation learner. Based on the generalizable embedding space, CLIP has been utilized on numerous tasks [58, 59, 60, 61], and there are many attempts to fine-tune it efficiently [62, 63, 64, 65]. Moreover, robust fine-tuning methods have also been actively studied for generalization on both in- and out-of-distribution data. Wortsman et al. [10] propose a weight-ensemble method that interpolates the pre-trained and fine-tuned weights, Kumar et al. [11] adopt a two-stage learning scheme that combines linear probing and fine-tuning, and Goyal et al. [66] adopt contrastive loss during fine-tuning on image classification. However, previous works have only focused on uni-modal settings and downstream tasks are restricted to the image classification. This paper provides robust fine-tuning methods for multi-modal settings.

## 3   Observations and Problem Define

For a given batch $D = \{x_i, y_i\}_{i=1}^M$ of $M$ instances where $(x_i, y_i)$ denotes the $i$-th image-text pair, the goal of multi-modal learning is to learn the pair-wise relations between image and text. To do this, CLIP [19] learns image encoder $f(\cdot; \theta_1)$ and text encoder $g(\cdot; \theta_2)$, so that embedding pairs $\{I_i, T_i\}_{i=1}^M = \{f(x_i; \theta_1), g(y_i; \theta_2)\}_{i=1}^M$ get closer to each other. Note that $I_i$ and $T_i$ are $L_2$-normalized unit vectors in most vision-language models, and they lie on the hypersphere. For simplicity, we omit the learning parameters $\theta_1$ and $\theta_2$ from all the following equations. CLIP adopts InfoNCE-style [13] loss $C(\cdot, \cdot)$ to enforce the similarity between positive pairs $(x_i, y_i)$ and distance among all remain negative pairs $(x_i, y_j)$. This is formulated as below (Eq. 1):

$$C(I, T) = \frac{1}{M} \sum_{i=1}^M - \log \frac{\exp(sim(I_i, T_i)/\tau)}{\sum_{j=1}^M \exp\left(sim(I_i, T_j)/\tau\right)} \qquad \mathcal{L}_{\text{CLIP}} = \frac{1}{2}(C(I, T) + C(T, I)) \quad (1)$$

Like many other CL approaches, CLIP uses a dot product as a similarity calculation $sim(\cdot, \cdot)$ between two vectors and governs $\tau$ as a *learnable* temperature that controls the scale of measured similarity. Now, we analyze the multi-modal embedding in terms of uniformity and alignment, well-known properties in uni-modal CL literature [25, 26] but unexplored in multi-modal settings. Alignment[1] (Eq. 2) evaluates the difference between distances (or similarities) of positive pairs compared with

---

[1]Original formulation of alignment is $\mathbb{E}_{(x_i, y_i)}[\|f(x_i) - g(y_i)\|_2^2]$, which ignores the similarity among negative pairs. We found that it is not directly related to downstream performances, so we modified the alignment to relative-alignment that handles the similarities of both positive and negative pairs.

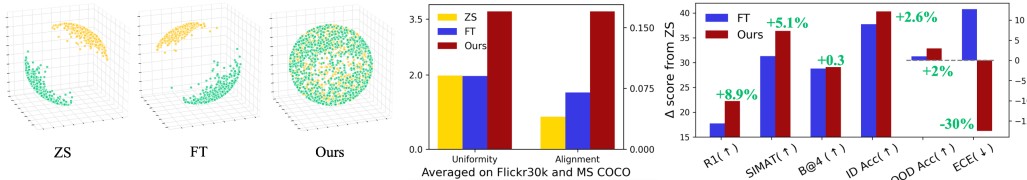

Figure 2: (Left) DOSNES [24] visualization of CLIP's embedding space on Flickr 30k [67]. Greens and oranges denote the image and text embedding, respectively. Embeddings of pre-trained (ZS) and naively fine-tuned (FT) ones have two separate clusters with low uniformity and alignment (Middle), which may limit embedding transferability. Multi-modal mixup induces more aligned and uniformly distributed embeddings, (Right) which largely improves downstream performance, including retrieval (R1), embedding arithmetic (SIMAT), image captioning (BLUE@4), and classification (ID Acc. and OOD Acc.), and uncertainty calibration (ECE).

the hardest negative pairs, while uniformity (Eq. 3) indicates how uniformly the data is distributed. The greater alignment and uniformity denote the more transferable representation [25, 26].

$$\text{Alignment} := -\mathbb{E}_{(x_i, y_i)} \left[ \|f(x_i) - g(y_i)\|_2^2 - \min_{k \neq i} \|f(x_i) - g(y_k)\|_2^2 \right] \quad (2)$$

$$\text{Uniformity} := -\log \mathbb{E}_{(x_i, y_j)} \left[ \exp \left( -2\|f(x_i) - g(y_j)\|_2^2 \right) \right] \quad (3)$$

Fig. 2 shows (Left) DOSNES [24] and (Middle) uniformity-alignment of CLIP's embedding space on image-caption datasets after pre-trained (ZS), naively fine-tuned (FT), and fine-tuned with our method (Ours). Pre-trained CLIP embeddings are separated by two subspaces for image and text with wide interspace between them, and even after fine-tuning, this structure remains unchanged. As a result, CLIP and its fine-tuned embedding have poor uniformity and alignment, and this might limit the transferability and robustness of embeddings to downstream data. From this observation, we aim to learn a more transferable and robust multi-modal representation with *multi-modal mixup*. To validate the transferability and robustness of embeddings, besides evaluating uniformity-alignment, we perform diverse downstream tasks: retrieval, few-/zero-shot classification on in-distribution (ID), out-of-distribution (OOD), classification under modality missing, embedding arithmetic, and image captioning in Section 5. Here, our method produces well-aligned and uniformly distributed embeddings resulting in consistent downstream performance gains (Fig. 2 Right[2]).

## 4 Methodology

### 4.1 Understanding Geodesic Multi-Modal Mixup

From our finding that CLIP embedding is not robust enough, we improve it by fine-tuning CLIP via contrastive loss with virtual hard negative samples. First, we generate hard negatives by mixing the image and text embeddings via **multi-modal Mixup**, $m^2$-Mix. Note that CLIP's $L_2$-normalized embeddings lie on a hypersphere, and the mixed ones are also desirable to lie on that hypersphere. However, original Mixup [37, 41] does not guarantee that mixed data lie on the hypersphere. Therefore, we devise a new type of Mixup, **geodesic Mixup** (defined as Eq. 4). Geodesic Mixup interpolates two data points on geodesic path, so it ensures that mixed samples lie on the hypersphere.

$$m_\lambda(\vec{a}, \vec{b}) = \vec{a} \frac{\sin(\lambda\theta)}{\sin(\theta)} + \vec{b} \frac{\sin((1-\lambda)\theta)}{\sin(\theta)}, \quad \text{where } \theta = \cos^{-1}(\vec{a} \cdot \vec{b}) \text{ and } \lambda \sim \text{Beta}(\alpha, \alpha) \quad (4)$$

Where $\lambda$ is handled by a hyperparameter $\alpha$ in Beta distribution. It is well-known that $L_2$-normalized embeddings are crucial for metric learning [68, 69] thus adopted by most of the modern contrastive learning methods [14, 15, 19]. Therefore, it is necessary that mixture samples lie on the unit sphere, which is guaranteed by our geodesic Mixup. Comparison between geodesic Mixup and standard Mixup following manual $L_2$-normalization in Tab. 6. Then, we utilize the generated hard negatives for contrastive loss by replacing the original negatives (See Eq. 5). Here, we only change the denominator term and retain the numerator that compares the similarity between the positive pairs.

$$C_{m^2}(I, T) = \frac{1}{M} \sum_{i=1}^{M} -\log \frac{\exp(I_i \cdot T_i/\tau)}{\exp(I_i \cdot T_i/\tau) + \sum_{j \neq i} \exp(I_i \cdot m_\lambda(I_j, T_j)/\tau)} \quad (5)$$

$$\mathcal{L}_{m^2\text{-Mix}} = \frac{1}{2}(C_{m^2}(I, T) + C_{m^2}(T, I))$$

---

[2]For each method, FT and Ours, we compute scores for each bar by averaging all values of FT and $m^3$-Mix in corresponding Tables, except OOD Acc computed by averaging values of WiSE-FT, LP-FT, and MaPLe.

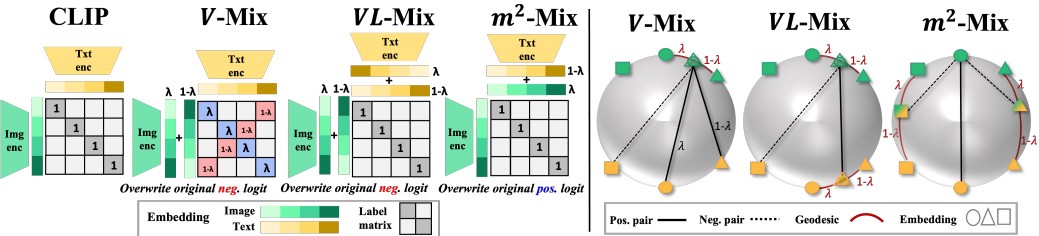

Figure 3: Comparison among contrastive losses. CLIP enforces the pair-wise similarity between matched image-text embedding pairs. $m^2$-Mix generates hard negative samples via mixing two heterogeneous embeddings. We additionally propose uni-modal Mixups, $V$-Mix (and $L$-Mix) and $VL$-Mix, that augment the homogenous embeddings in multi-modal contrastive loss.

Because CLIP has two-sided polarized embeddings, the similarity between the original image (or text) embedding and mixed embedding is larger than that between the original image and text embeddings. Therefore, $m^2$-Mix generates harder negative samples compared with original negatives. The existence of hard negative samples is known to be important in uni-modal CLs [25, 26], and we validate it under multi-modal CL settings in Section 5. Besides, we provide a theoretical result on hardness guarantee and limiting behavior of $\mathcal{L}_{m^2\text{-Mix}}$ in the following paragraph.

**Theoretical Analysis**  In Section 3, we observed that naive fine-tuning with standard contrastive loss could not reduce the embedding separation. We speculate this limitation is derived from (1) lack of hard negative samples and (2) vanished learnable $\tau$ (0.01) in $\mathcal{L}_{\text{CLIP}}$. As shown by Wang et al. [26], when $\tau$ approaches zero, the contrastive loss behaves like *triplet loss* with zero-margin. This means that if the similarity between positive samples is greater than that of the nearest negative, there are no incentives to further pull or push the embeddings of positive and negative samples. Moreover, due to CLIP's bipartite embedding space, it might lack sufficient hard negative samples to incentivize the models to pursue more alignment and uniformity. Therefore, we argue that **hard negatives are necessary for multi-modal CL** when there is an embedding space modality gap [27].

**Theorem 4.1** (Hardness of $m^2$-Mixed samples). *Let's assume that two random variables $x_1$ and $x_2$ follow the $M_d(\mu_1, \kappa)$ and $M_d(\mu_2, \kappa)$, von Mises–Fisher distribution with mean direction $\mu_1, \mu_2$ and concentration parameter $\kappa$ in $\mathbb{R}^d$, respectively. Let $\widetilde{x} = x_1 + x_2$ and $d = 2$. Then, $D_{KL}(p(x_1)||p(\widetilde{x})) \leq D_{KL}(p(x_1)||p(x_2))$ for sufficiently large $\kappa$.*

Theorem 4.1 shows that KL-divergence between the pair of an original sample and a mixed sample is less (more confused with positive) than that of another original sample (proof in Supp. D). Meanwhile, because the converged $\tau$ of CLIP is significantly small (i.e., 0.01), it will be reasonable to consider an extreme case: when $\tau \to 0^+$. Based on Proposition 4.2, we argue that ones can explicitly enforce uniformity-alignment in multi-modal contrastive learning by equipping $m^2$-Mix with $\mathcal{L}_{\text{CLIP}}$.

**Proposition 4.2** (Limiting behavior of $\mathcal{L}_{\text{CLIP}}$ with $\mathcal{L}_{m^2\text{-Mix}}$). *For sufficiently large $M$, as the temperature of contrastive loss $\tau \to 0^+$, the $\mathcal{L}_{CLIP}$ and $\mathcal{L}_{m^2\text{-Mix}}$ converges to the triplet loss with zero-margin (i.e., corresponding to negative Alignment) and negative Uniformity, respectively. That is: $\lim_{\tau \to 0^+} \mathcal{L}_{CLIP} + \mathcal{L}_{m^2\text{-Mix}} \simeq -(Alignment + Uniformity)$*

$m^2$-Mix brings two advantages on multi-modal CL with theoretical grounds. Firstly, by generating sufficiently hard negatives, it incentivizes the model to enforce alignment more strongly whether there exists a modality gap or not. Besides, the uniformity is also explicitly increased as $\mathcal{L}_{m^2\text{-Mix}}$ asymptotically converges to negative uniformity. Thus, $\mathcal{L}_{\text{CLIP}}$ with $m^2$-Mix induces well-aligned and uniformly distributed embeddings, so it makes the model robustly works on diverse tasks.

## 4.2 Uni-Modal Mixup for CLIP

$m^2$-Mix generates hard negatives for CL to align and distribute embeddings uniformly. Meanwhile, Mixup is known to alleviate the overconfidence problem in uni-modal setups [38]. So, we further propose three uni-modal Mixups, Vision-Mix ($V$-Mix), Language-Mix ($L$-Mix), and Vision-Language Mix ($VL$-Mix) to enhance multi-modal CL. Fig. 3 shows the overall structures.

*uni*-**Mix**  $V$-Mix and $L$-Mix interpolate the image and text embedding, respectively. To be specific, $V$-Mix mixes the embeddings of images in batch and a flipped (reversed) batch with a ratio $\lambda$. Then,

$m_\lambda(I_i, I_{i'})$ has information from $i$ and $i' = M - i$ indexed samples with $\lambda$ and $1 - \lambda$ fraction, respectively. Thus, pseudo label for $(m_\lambda(I_i, I_{i'}), T_i)$ pair is $\lambda$ while, that for $(m_\lambda(I_i, I_{i'}), T_{i'})$ is $1$-$\lambda$.

$$C_V(I, T) = \frac{1}{M} \sum_{i=1}^{M} -\lambda \log \frac{\exp(m_\lambda(I_i, I_{i'}) \cdot T_i / \tau)}{\sum_{j=1}^{M} \exp(I_i \cdot T_j / \tau)} - (1 - \lambda) \log \frac{\exp(m_\lambda(I_i, I_{i'}) \cdot T_{i'} / \tau)}{\sum_{j=1}^{M} \exp(I_i \cdot T_j / \tau)}$$

$L$-Mix has the same formula with $V$-Mix, except that it is applied to text-side. The combination loss term of $V$- and $L$-Mix is defined as $\mathcal{L}_{uni\text{-Mix}} = \frac{1}{2}(C_V(I, T) + C_V(T, I)) + \frac{1}{2}(C_L(I, T) + C_L(T, I))$

$VL$**-Mix**  For pair-wise similarity contrast, $V$-Mix and $L$-Mix only mix the image and text embedding, respectively. We additionally propose $VL$-Mix that mixes the image and text embedding simultaneously. Note that $m^2$-Mix mixes embeddings of image and text, while $VL$-Mix independently mixes them. Both $m_\lambda(I_i, I_{i'})$ and $m_\lambda(T_i, T_{i'})$ has $i$-th component and $i'$-th component with fraction $\lambda$ and $1 - \lambda$ respectively, so the pseudo label for $(m_\lambda(I_i, I_{i'}), m_\lambda(T_i, T_{i'}))$ is 1. Here, similarities between negative pairs are retain with that of original negatives likewise $uni$-Mix.

$$C_{VL}(I, T) = \frac{1}{M} \sum_{i=1}^{M} - \log \frac{\exp(m_\lambda(I_i, I_{i'}) \cdot m_\lambda(T_i, T_{i'}) / \tau)}{\sum_{j=1}^{M} \exp(I_i \cdot T_j / \tau)} \quad \mathcal{L}_{VL\text{-Mix}} = \frac{1}{2}(C_{VL}(I, T) + C_{VL}(T, I))$$

$m^3$**-Mix**  We name the combination of $m^2$-Mix with uni-Mix and $VL$-mix as $m^3$-Mix, multiple multi-modal Mixup. Complete objective function is denoted as (weights for each term are omitted):

$$\mathcal{L}_{m^3\text{-Mix}} = \mathcal{L}_{CLIP} + \mathcal{L}_{m^2\text{-Mix}} + \mathcal{L}_{uni\text{-Mix}} + \mathcal{L}_{VL\text{-Mix}}$$

## 5  Results

**Settings**  Unless otherwise stated, we adopt CLIP ViT-B/32 as our backbone model. We consider the following methods as our baselines in retrieval and embedding arithmetic tasks: zero-shot inference (ZS) of CLIP, embedding shift (ES) [27], naive fine-tuning (FT), and its increased $\tau$ variants, recent Mixup-augmented uni-modal CL methods $i$-Mix [50] and Un-Mix [51]. We put task-specific setups on each section and Sec. A of Supplementary Material (SM). Further details, hyperparameters selection, pseudo code, and additional results are put in Sec. A, B, and C of SM, respectively.

### 5.1  Generated Samples by $m^2$-Mix

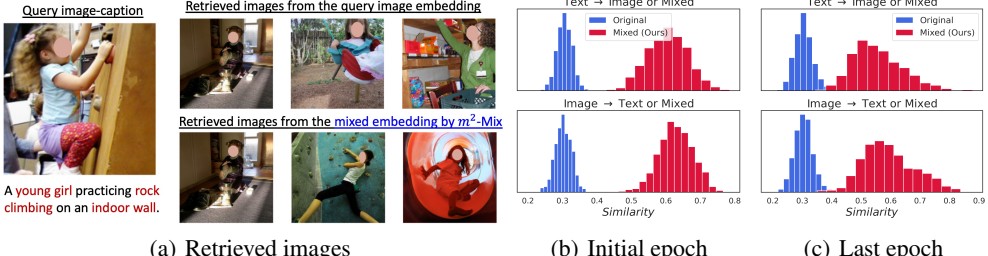

(a) Retrieved images          (b) Initial epoch          (c) Last epoch

Figure 4: On Flickr30k, (a) top-3 retrieved images by image embedding (Top) and $m^2$-Mixed (Bottom) from a source instance (Left). $m^2$-Mix generates an embedding that contains features from both modalities (young girl, climbing, and indoor wall), partly lacking in the image embedding. (b) and (c) denote cosine similarities between given test instances and its top-1 nearest negative embedding during training epochs. $m^2$-Mix makes negatives that are highly similar to given instances.

To understand the properties of generated embedding by $m^2$-Mix, we explore the mixed embedding from $m^2$-Mix. Due to the non-trivial visualization of the embedding itself, we retrieve images that have the most similar embedding with a mixed one. In Fig. 4 (a), the embedding generated by $m^2$-Mix has both features from image and text that lack in original image embedding, e.g., the second and third images from $m^2$-Mix have rock climbing and indoor wall represented in the text. Besides, similarity histograms in Fig. 4 (b) and (c), show that $m^2$-Mix consistently produces harder negatives than the original non-mixed counterparts from the initial to last training epochs.

## 5.2 Cross-Modal Retrieval with CLIP

First, we validate our method on image-text retrieval, a representative vision-language task, on Flickr30k [67] and MS COCO [70]. All methods are trained over 9 epochs with Adam optimizer (details in SM). Tab. 1 denotes top-1/-5 recall of retrieval. Our $m^3$-Mix increases overall performance, while the standard fine-tuning approaches and Mixup-baselines [50, 51] have limited performance gain. Corresponding to previous works [26, 27], we also found that properly increasing temperature ($\tau$) in contrastive loss is quite beneficial at improving performance for both FT and $m^3$-Mix.

Table 1: Image to text (i→t) and text to image retrieval (t→i) retrieval results (top-1/-5 Recall;R1, R5). ZS and FT denote pre-trained and fine-tuned CLIP, respectively.

| | Flickr30k | | | | MS COCO | | | |
| | i→t | | t→i | | i→t | | t→i | |
| | R1 | R5 | R1 | R5 | R1 | R5 | R1 | R5 |
|---|---|---|---|---|---|---|---|---|
| ZS | 71.1 | 90.4 | 68.5 | 88.9 | 31.9 | 56.9 | 28.5 | 53.1 |
| ES [27] | 71.8 | 90.0 | 68.5 | 88.9 | 31.9 | 56.9 | 28.7 | 53.0 |
| FT | 81.2 | 95.4 | 80.7 | 95.8 | 36.7 | 63.6 | 36.9 | 63.9 |
| FT ($\tau = 0.05$) | 82.4 | 95.1 | 82.1 | 95.7 | 40.2 | 68.2 | 41.6 | **69.9** |
| FT ($\tau = 0.10$) | 75.7 | 93.9 | 78.0 | 92.9 | 34.2 | 62.7 | 36.7 | 64.2 |
| $i$-Mix [50] | 72.3 | 91.7 | 69.0 | 91.1 | 34.0 | 63.0 | 34.6 | 62.2 |
| Un-Mix [51] | 78.5 | 95.4 | 74.1 | 91.8 | 38.8 | 66.2 | 33.4 | 61.0 |
| $m^3$-Mix | 82.3 | **95.9** | 82.7 | **96.0** | **41.0** | **68.3** | 39.9 | 67.9 |
| $m^3$-Mix ($\tau = 0.05$) | **82.7** | 95.7 | **82.8** | 95.5 | 40.4 | 67.9 | **42.0** | 68.8 |

Table 2: Retrieval with disjoint unimodal models. ZS is a naive combination of two models without joint-tuning.

| | Flickr30k | | | |
| | i → t | | t → i | |
| | R1 | R5 | R1 | R5 |
|---|---|---|---|---|
| ZS | 0.1 | 0.4 | 0.1 | 0.2 |
| ES [27] | 0.1 | 0.5 | 0.2 | 0.2 |
| FT | 28.7 | 61.7 | 26.7 | 59.4 |
| FT ($\tau = 0.05$) | 31.5 | 64.2 | 29.2 | 61.4 |
| FT ($\tau = 0.10$) | 30.0 | 62.7 | 30.1 | 60.6 |
| $i$-Mix [50] | 27.6 | 60.3 | 27.1 | 60.7 |
| Un-Mix [51] | 31.5 | 64.3 | 29.2 | 61.2 |
| $m^3$-Mix | 31.9 | 62.6 | 30.3 | 61.0 |
| $m^3$-Mix ($\tau = 0.05$) | **32.5** | **64.7** | **30.4** | **63.4** |

Besides, we observed the improved uniformity and alignment by $m^3$-Mix (Fig. 2) not only enhances the Recall of retrievals but also contributes to the calibration [71]. The left side of Fig. 5 denotes the reliability diagrams with calibration errors of the text-to-image retrieval R1 score on Flickr30K. While the naively fine-tuned CLIP has poor calibration, fine-tuning with $m^3$-Mix alleviates the overconfidence issue somewhat and results in a better calibration. This is further confirmed by Tab. 3, in which our $m^3$-Mix significantly improves the Expected Calibration Error (ECE) of CLIP. Meanwhile, it is known that the ECE value can be improved by adjusting the temperature $\tau$, i.e., temperature scaling [71]. Therefore, we provide the sensitive analysis on varying $\tau$. In Fig. 5 right side, our method shows relatively robust ECE under varying $\tau$, implying that our multi-modal Mixup-based CL induces the well-calibrated multi-modal model, which is crucial for reliable AI applications.

Table 3: Calibration on Flickr30k.

| Metric | Task | ZS | FT | $m^3$-Mix |
|---|---|---|---|---|
| ECE ($\downarrow$) | i → t | 1.90 | 2.26 | **1.54** |
| | t → i | 1.88 | 2.00 | **1.58** |

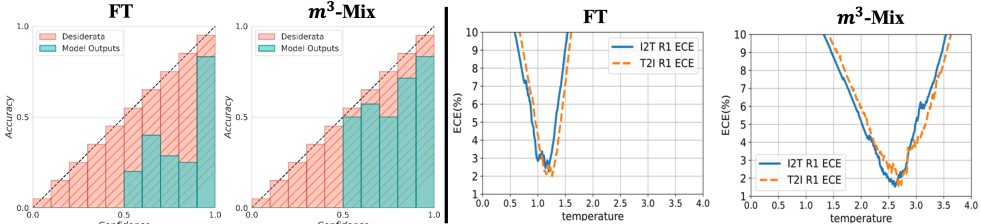

Figure 5: Reliability diagram (left) and ECE under varying temperature $\tau$ (right) on Flickr30K image-to-text retrieval. Our method shows (1) a better reliability diagram (close to $y = x$), (2) achieves a lower minimum ECE value, and (3) more stable across varying $\tau$ than naive fine-tuning. Thus, representation learning by $m^3$-Mix robustly induces a well-calibrated multi-modal model.

## 5.3 Cross-Modal Retrieval with Uni-Modal Pre-Trained Models

Sometimes, the high-quality annotations for multi-modal datasets are expensive, and there are cases when plenty of paired multi-modal data is unavailable. Then, it is crucial to exploit the uni-modal pre-trained models for learning multi-modal embedding space [72]. To this end, we validate our $m^3$-Mix on the fine-tuning of disjointly pre-trained uni-modal models. Specifically, we jointly fine-tune the pre-trained BERT [73] and ResNet-50 [74] with a contrastive loss on Flickr30k (in Table 2). Among candidates, $m^3$-Mix with higher $\tau$ consistently achieves the highest performance so that it can be adopted as an effective joint tuning method for independently pre-trained uni-modal models.

## 5.4 Few-Shot Adaptation and Robustness on Distribution Shift

Table 4: Few-shot adaptation under general setting.

| Method | Dataset | | | |
|--------|---------|------|-------|------|
| | Pets | SVHN | CLEVR | Avg. |
| ZS | 87.49 | 13.63 | 20.70 | 40.61 |
| FT | 89.37 | 45.00 | 53.49 | 62.62 |
| FT w/ $V$-Mix | 89.45 | 44.61 | 53.93 | 62.66 |
| FT w/ $L$-Mix | 89.43 | 48.42 | 53.91 | 63.92 |
| FT w/ $VL$-Mix | 89.56 | 45.22 | 53.75 | 62.84 |
| FT w/ $m^2$-Mix | 90.05 | 46.24 | 53.60 | 63.29 |
| $m^3$-Mix | 90.16 | 54.84 | 53.85 | 66.28 |
| $m^3$-Mix ($\tau = 0.05$) | 90.49 | 60.90 | 53.95 | 68.45 |
| WiSE-FT [10] | 91.80 | 35.04 | 41.93 | 56.25 |
| WiSE-FT w/ $m^3$-Mix | **92.51** | 58.55 | 47.11 | 66.06 |
| LP-FT [11] | 89.92 | 44.91 | 53.62 | 62.82 |
| LP-FT w/ $m^3$-Mix | 91.03 | **64.24** | **55.20** | **70.16** |
| MaPLe [64] | 90.87 | 47.62 | 43.05 | 60.51 |
| MaPLe w/ $m^3$-Mix | 91.14 | 52.72 | 45.20 | 63.02 |

Table 5: Few-shot (ImageNet; IN) and zero-shot evaluation under distribution shift (-V2, -A, -R, -S).

| Method | Dataset | | | | | |
|--------|---------|-------|------|------|------|------|
| | IN | IN-V2 | IN-A | IN-R | IN-S | Avg. |
| ZS | 62.06 | 54.80 | 29.63 | 66.02 | 40.82 | 50.67 |
| FT | 65.44 | 55.35 | 20.07 | 58.16 | 34.50 | 46.70 |
| FT w/ $V$-Mix | 66.00 | 56.19 | 20.85 | 60.50 | 34.97 | 47.70 |
| FT w/ $L$-Mix | 65.96 | 55.95 | 20.57 | 60.54 | 35.25 | 47.65 |
| FT w/ $VL$-Mix | 66.24 | 56.70 | 21.36 | 61.07 | 35.11 | 48.10 |
| FT w/ $m^2$-Mix | 67.04 | 57.39 | 20.05 | 59.28 | 35.31 | 47.81 |
| $m^3$-Mix | 67.08 | 57.55 | 20.80 | 60.96 | 35.86 | 48.45 |
| $m^3$-Mix ($\tau = 0.05$) | 68.40 | 58.51 | 22.17 | 62.28 | 37.62 | 49.80 |
| WiSE-FT [10] | 69.00 | 59.66 | 28.01 | 64.84 | 41.05 | 52.51 |
| WiSE-FT w/ $m^3$-Mix | **69.65** | **60.71** | 29.16 | 66.75 | 42.19 | **53.69** |
| LP-FT [11] | 68.22 | 58.40 | 25.57 | 63.36 | 38.04 | 50.72 |
| LP-FT w/ $m^3$-Mix | 68.62 | 59.17 | 25.85 | 65.14 | 38.78 | 51.51 |
| MaPLe [64] | 65.59 | 58.44 | 32.49 | 68.13 | 42.53 | 53.44 |
| MaPLe w/ $m^3$-Mix | 65.76 | 58.16 | **32.52** | **68.20** | **42.67** | 53.46 |

Next, we evaluate our methods on few-shot image classification under general (in Tab. 4) and distribution shift settings (in Tab. 5 and 6). We consider OxfordPets [75], SVHN [76], and CLEVR [77] for the general setting[3] and ImageNet-1k, ImageNetV2 [78], ImageNet-A [79], ImageNet-R [80], and ImageNet-Sketch [81] for distribution shift setting. Unlike MS COCO and Flickr30K, These datasets provide class name labels only and do not have captions corresponding to each image. To make CL methods amendable for this setting, we adopt a common prompt 'a photo of classname' that wraps the class name with a short context and use this as captions of images. Following [82, 64], we perform the tasks under a few-shot evaluation protocol: 16-shot training samples per class and inference on the entire test set. As baselines, we first consider zero-shot CLIP (ZS) and construct the contrastive loss adoption of vanilla fine-tuning (FT). Then, we showcase our methods with exhaustive ablation ($V$-, $L$-, $VL$-, and $m^2$-Mix) as well as our complete objective $m^3$-Mix with its high-temperature variant. To further compare our approach with state-of-the-art (SOTA) fine-tuning methods, we consider MaPLe [64] that optimizes the continuous prompts inside the text and image encoders of CLIP, and the contrastive loss extended version of uni-modal fine-tune methods: LP-FT [11] and WiSE-FT [10].

In both general and distribution shift settings, $m^2$-Mix and uni-modal Mixups ($V$-, $L$-, $VL$-) contribute to boost the few-/zero-shot classification performance. After integrating them, $m^3$-Mix and its high-temperature variant give significant performance improvement, implying $m^3$-Mix is an effective fine-tuning method that covers challenge generalization setup. Moreover, when $m^3$-Mix combined with SOTA fine-tuning methods [10, 11, 64],

Table 6: Ablation study on Mixup.

| Temperature ($\tau$) | $m^3$-Mix type | |
|----------------------|--------|----------|
| | linear | geodesic |
| 0.01 | 48.36 | **48.45** |
| 0.05 | 48.48 | **49.80** |
| 0.10 | 45.20 | **46.41** |

it consistently brings performance. Therefore, $m^3$-Mix is a flexible plug-in method that can be collaborated with many algorithms. Besides, in Tab. 6 (mean Acc. of ImageNet variants), we show that geodesic Mixup achieves superior results than linear Mixup with manual $L_2$-normalization. Thus, based on its analytic property, geodesic Mixup is more suitable than linear Mixup under frameworks that learn representation on a hypersphere, as in modern CLs.

## 5.5 Robustness on Modality Missing

In this section, we study whether $m^2$-Mix can help the multi-modal representation learning for video recognition (CMU-MOSEI [83]) under modality missing. Recently, Geometric Multi-modal Contrastive Learning (GMC) [84] achieved competitive results on CMU-MOSEI.

However, GMC only considers the uni-to-joint modal relationship [84], while our $m^2$-Mix explicitly generates the mixture of bi-modal semantics so that it can additionally consider the bi-to-joint modal relation. From this, we hypothesize that $m^2$-Mix can further improve the robustness and informativeness of the representation. For evaluation, we add the $m^2$-Mix on top of GMC.

---

[3]Results of CLIP ViT-B/16 on other transfer learning benchmark datasets are provided in Sec. C of SM.

Different from the CLIP fine-tuning cases, we use the multi-modal mixed representation as both positive and negative pairs with the target joint representation because our goal in this task is to align the embedding between the joint and other modalities. As shown in Tab. 7, $m^2$-Mix coherently improves the performance of GMC in terms of accuracy, alignment, and uniformity. While GMC strongly aligns the embedding of partial and joint modality, the alignment is further enhanced by the aid of $m^2$-Mix (also confirmed in Fig. 6), which results in superior performance when only partial information is given in test-time (modality missing).

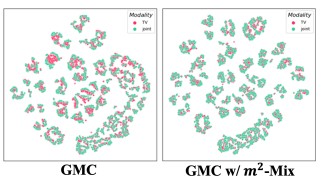

**GMC**      **GMC w/ $m^2$-Mix**

Figure 6: t-SNE [85] on CMU-MOSEI with partial information. Each color denotes embeddings of partial and joint modality.

These results justify the use of $m^2$-Mix for robust learning under modality-incomplete scenarios.

Table 7: Accuracy (acc.)(↑), alignment (align.)(↑), and uniformity (unif.)(↑) of multi-modal learning methods on CMU-MOSEI under complete and partial modalities. Averaged performance of five runs.

| | | | | | | | | | | | | | | | | | | | | |
|---|---|---|---|---|---|---|---|---|---|---|---|---|---|---|---|---|---|---|---|---|
| | | | | | | | | Test-time Observed Modalities | | | | | | | | | | | | |
| | Full (T+V+A) | | T | | | V | | | A | | | T+V | | | T+A | | | V+A | | |
| | acc. | unif. | acc. | align. | unif. | acc. | align. | unif. | acc. | align. | unif. | acc. | align. | unif. | acc. | align. | unif. | acc. | align. | unif. |
| MulT [86] | 80.5 | 0.99 | 60.0 | - | 1.03 | 53.9 | - | 2.07 | 52.7 | - | 0.62 | 57.8 | - | 1.27 | 58.8 | - | 0.77 | 54.6 | - | 1.36 |
| GMC [84] | 80.1 | 3.06 | 78.5 | 0.20 | 3.03 | **64.7** | 0.17 | 3.01 | 66.0 | 0.09 | 3.03 | 77.0 | 0.07 | 2.94 | 77.4 | 0.08 | 3.00 | 67.3 | 0.05 | 2.98 |
| GMC+$m^2$-Mix | **80.5** | **3.18** | **78.9** | **0.23** | **3.17** | 64.2 | **0.19** | **3.15** | **66.2** | **0.12** | **3.15** | **77.8** | **0.08** | **3.08** | **77.9** | **0.09** | **3.08** | **67.4** | 0.06 | **3.10** |

## 5.6 Multi-Modal Embedding Arithmetic

We expect that well-learned multi-modal embedding represents the structural relationship between instances like word vectors [87]. For validation, we evaluate the learned embeddings on SIMAT [88]. SIMAT evaluates text-driven image representation by retrieving a new image with the highest similarity to the latent vector $x$, which is transformed by text *delta vectors* when we change the word in an original text, i.e., formulated as: $x = I_{original} + \lambda \cdot (T_{new} - T_{original})$. Here, $I$ and $T$ are image and text embedding vectors, and $\lambda$ is a hyper-parameter about the strength of transformation. Table 8 presents the quantitative scores across methods after fine-tuning on Flickr30k and MS COCO, and evaluated on SIMAT. Learned representation from $m^3$-Mix shows stable scores on both a multi-modal model and jointly fine-tuned uni-modal models, which is further confirmed by qualitative experiments (Fig. 7). These support that $m^3$-Mix can be adopted as a delicate fine-tuner when embedding geometric structure and arithmetic property should be considered, e.g., controllable generation [89].

Table 8: $m^3$-Mix shows the stable SIMAT Score (↑) on CLIP and joint fine-tuning of uni-modal pre-trained models.

| | CLIP (MS COCO) | CLIP (Flickr30k) | BERT + RN50 (Flickr30k) |
|---|---|---|---|
| ZS | 34.5 | 34.4 | 6.1 |
| ES [27] | 34.6 | 34.5 | 6.2 |
| FT | 42.3 | **40.8** | 15.4 |
| $i$-Mix [50] | 37.3 | 40.0 | 12.9 |
| Un-Mix [51] | 42.9 | 38.5 | 15.8 |
| $m^3$-Mix | **44.4** | 38.9 | **19.0** |

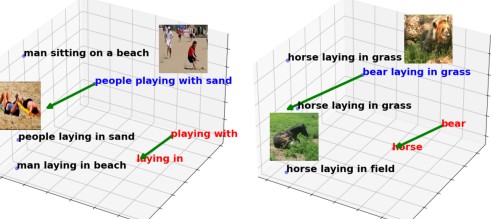

Figure 7: Embeddings of $m^3$-Mix on SIMAT source-target texts (reds) and images (source and top-1 retrieved). Black texts correspond to top-3 retrieved images from a source image-caption.

## 5.7 Multi-Modal Mixup on a State-of-the-Art Vision-Language Model

In this section, we investigate whether our multi-modal Mixup can also be beneficial for improving other recent large-scale vision-language models beyond CLIP. For this, we adopt Contrastive Captioner (CoCa) [90] ViT-L/14 configuration that pre-trained on LAION-2B [5] from `OpenCLIP` library as our target backbone model. We consider three learning objectives for CoCa fine-tuning: (1) autoregressive captioning loss (Cap), (2) contrastive loss and captioning loss (CL + Cap), and (3) contrastive loss, $\mathcal{L}_{m^2\text{-Mix}}$, and captioning loss (CL w/ $\mathcal{L}_{m^2\text{-Mix}}$ + Cap). For all three methods, we train the model on MS COCO over one epoch with `OpenCLIP`-provided hyperparameter configuration. After that, we evaluate each model for image captioning (Tab. 9) on MS COCO and zero-shot and fine-tuned cross-modal retrieval (Tab. 10) on Flickr30K and MS COCO.

Table 9: Image captioning results on MS COCO with CoCa ViT-L/14 model. We fine-tune the CoCa on MS COCO for 1 epoch with three different learning objectives and evaluate them in terms of five conventional metrics. $m^2$-Mix achieves performance gain on all the five metrics.

| Method | Metrics | | | | |
|--------|---------|--------|---------|-------|-------|
| | BLEU@4 | METEOR | ROUGE-L | CIDEr | SPICE |
| ZS | 7.2 | 12.4 | 26.3 | 35.2 | 9.3 |
| Cap | 36.0 | 29.4 | 57.3 | 125.1 | 23.1 |
| CL + Cap | 35.7 | 29.3 | 57.1 | 124.9 | 23.0 |
| CL w/ $\mathcal{L}_{m^2\text{-Mix}}$ + Cap | **36.3** | **29.5** | **57.5** | **125.6** | **23.2** |

In Tab. 9, while a combination of vanilla contrastive loss with captioning loss underperforms the captioning-loss-only training, $m^2$-Mix-assisted contrastive learning further increases the performance on image captioning in terms of five conventional metrics. Besides, Tab. 10 shows that $m^2$-Mix generally improves the retrieval recalls of CoCa on zero-shot and fine-tuned settings. This implies that representation learning with our multi-modal Mixup is beneficial to generative tasks as well as discriminative tasks. The consistent improvement shown in the image captioning task is accorded with observations from other recent works [52, 53] that reveal the effectiveness of cross-modal Mixup on generative tasks by increasing cross-modal alignment.

Table 10: Cross-modal retrieval top-1/5 recalls on MS COCO (fine-tuned) and Flickr30k (zero-shot transfer). $m^2$-Mix generally enhances image-text retrieval of a SOTA vision-language model, CoCa.

| Method | (Zero-shot) Flickr30k | | | |
|--------|------------------|------------------|------------------|------------------|
| | i $\rightarrow$ t (R1) | i $\rightarrow$ t (R5) | t $\rightarrow$ i (R1) | t $\rightarrow$ i (R5) |
| Cap | 90.4 | 98.5 | 78.5 | 94.4 |
| CL + Cap | 92.4 | **99.1** | 79.2 | 94.9 |
| CL w/ $\mathcal{L}_{m^2\text{-Mix}}$ + Cap | **92.8** | 99.0 | **79.5** | **95.1** |
| Method | (Fine-tuned) MS COCO | | | |
| | i $\rightarrow$ t (R1) | i $\rightarrow$ t (R5) | t $\rightarrow$ i (R1) | t $\rightarrow$ i (R5) |
| Cap | 68.5 | 87.9 | 53.2 | 77.8 |
| CL + Cap | **73.9** | **91.2** | 56.3 | 80.4 |
| CL w/ $\mathcal{L}_{m^2\text{-Mix}}$ + Cap | **73.9** | 91.0 | **56.5** | **80.6** |

## 6  Conclusion

This paper analyzes the representation learned by a multi-modal contrastive learner. We found that CLIP has separated text-versus-image embedding space with poor uniformity-alignment. These polarized embeddings with huge unexploited space may limit the transferability and robustness of representation on downstream tasks. From our findings, we propose *Geodesic Multi-Modal Mixup* that generates hard negatives for robust contrastive learning by mixing two heterogeneous embeddings. Theoretically, we validate that our method produces hardness-guaranteed samples and has desirable asymptotic behavior to induce better generalizable representation. Empirically, the proposed method effectively improves performances on diverse tasks and perspectives: retrieval, calibration, few-shot classification under distribution shift, embedding arithmetic, and image captioning.

Though the increased uniformity and alignment of multi-modal representation largely empowers the model to adapt robustly to a diverse range of tasks, we found that reckless uplift of them is harmful in some cases of retrieval (modest increment of uniformity-alignment was better than huge increment of them). Thus, more research on the reliable evaluation of multi-modal representation should be pursued in the era of foundation models.

**Acknowledgement**  This work was supported by the National Research Foundation of Korea (NRF) grant funded by the Korea government (MSIT) (No.2021R1F1A1060117 and No.2022R1A4A3033874), and also supported by a grant (22183MFDS431) from Ministry of Food and Drug Safety in 2023.

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
