# A  Experiment Setup

We first elaborate on the setups for all the experiments, including retrieval and embedding arithmetic, uni-modal classification, multi-modal classification, and Contrastive Captioner (CoCa) image captioning and retrieval, in each separate section.

## A.1  Cross-modal Retrieval and SIMAT

**Dataset**    Here, we hold the Flickr30k and MS COCO, two representative vision-language benchmark datasets. Flickr30k contains 30K image-text pairs as a train split[4], 1k for validation and test splits[5]. For MS COCO, we adopt the 2017 version of it from the COCO Database[6]. MS COCO contains 118k image-text pairs for train split and 5k for both validation and test splits. When there are multiple captions for one image, we always select the first caption to construct an image-text pair. To validate the multi-modal embedding arithmetic, we use the SIMAT dataset [88]. SIMAT is a benchmark created for evaluating the text-driven image transformation performance of multi-modal embedding. It contains 6k images, 18k transformation queries that have pairs of (source word, target word, source image, target image), and 645 captions constructed with subject-relation-object triplets that have at least two corresponding images. The goal of SIMAT task is to retrieve an image, which is well-modified by a specific text transform to match with the ground truth transform target images.

**Model Description**    For retrieval and embedding arithmetic tasks, we adopt CLIP ViT-B/32 checkpoint of OpenAI official lease[7] as our backbone model. For cross-modal retrieval with disjointly pretrained uni-modal models, we utilize ResNet-50 [74] with a pre-trained checkpoint of `torchvision` as an image encoder and `BERT-base-uncased` from HuggingFace as a text encoder. To match the dimensions of these two uni-modal models, we add a projection head on top of each encoder, respectively.

**Baseline Methods**    First, we consider the zero-shot inference of CLIP (ZS) [19] as a strong baseline (in the case of retrieval with uni-modal pre-trained models, we just project the image and text embeddings to shared vector space with randomly initialized matrix, and perform similarity-based inference as ZS.), and embedding shift (ES) [27] which computes a delta vector (difference between mean vectors of image and text embeddings) and then manually modifies the modality gap along with delta vector direction without explicit training. Then, a vanilla fine-tuning (FT) with standard contrastive loss (Eq. 1 of main paper) and its higher-temperature variants ($\tau = \{0.05, 0.01\}$) are considered. Additionally, we take account of two uni-modal mixup-based contrastive learning methods $i$-Mix [50] and Un-Mix [51] those mix images in the input space. While the original implementation of $i$-Mix takes a randomly sampled image as a mixture component, we take a flipped batch sample as a mixture component for computational efficiency like as Un-Mix. So the only difference between $i$-Mix and Un-Mix is whether we construct the final objective as a sum of normal and mixed sample contrastive loss [51] or sorely mixed sample contrastive loss ([50]).

**Metric**    As a standard metric for retrieval tasks, we report top-1 recall (R1) and top-5 recall (R5) on both image-to-text and text-to-image directions. For SIMAT task, following the original paper [88], we performed the OSCAR-based evaluation and reported the SIMAT score in the original paper. It measures the similarity between the transformed image and text captions via OSCAR framework [91].

**Implementation Detail**    We fine-tune CLIP with `eval()` mode stable training[8] and under FP16 precision for computational efficiency. On both Flickr30k and MS COCO, we train each method over 9 epochs with batch size 128 via Adam optimizer ($\beta_1 = 0.9$, $\beta_2 = 0.98$, and $\epsilon = 1e-6$). As shared hyperparameters, we search for the best initial learning rate from {1e-6, 3e-6, 5e-6, 7e-6, 1e-5} and weight decay from {1e-2, 2e-2, 5e-2, 1e-1, 2e-1} for all training methods (Initial learning rate is decayed in each epoch by the exponential scheduler with decaying parameter 0.9). To construct our complete objective $m^3$-Mix, we weighing the $L_{\text{CLIP}}$ and $L_{m^2\text{-Mix}}$ and uni-modal geodesic Mixup

---

[4]https://www.kaggle.com/hsankesara/flickr-image-dataset

[5]https://github.com/BryanPlummer/flickr30k_entities

[6]https://cocodataset.org/#download

[7]https://github.com/openai/CLIP

[8]https://github.com/openai/CLIP/issues/150/

variants ($L_{V/L/VL\text{-Mix}}$). Specifically, we pivot the weight of $L_{\text{CLIP}}$ as 1.0 and sweep the weighting coefficient of other loss components for each dataset generally from $\{0.0, 0.01, 0.1, 0.2, 0.3, 0.5\}$[9]. The parameter $\alpha$ of Beta distribution $Beta(\alpha, \alpha)$ that determines the mixture ratio is set to 0.5 for the multi-modal Mixup and 2.0 for uni-modal Mixups. For embedding shift (ES) [27], we sweep $\lambda$ from -0.1 to 0.1 by 0.01 and report the best results among them. While the search range of ES from official implementation is from -2.5 to 2.5 by 0.125, we observe the finer search range gives better results.

## A.2 Uni-modal Classification

**Dataset**  We consider three common transfer learning benchmark datasets, OxfordPets [75], SVHN [76], and CLEVR [77], to validate the general few-shot adaptation capability. For evaluation of robustness on distribution shift, we consider the ImageNet-1k as a source dataset (models are adapted to) and ImageNetV2 [78], ImageNet-A [79], ImageNet-R [80], and ImageNet-Sketch [81] as target evaluation datasets those are considered as different kinds of natural distribution shift from ImageNet.

**Model Description**  For uni-modal few/zero-shot classifications, we also adopt CLIP ViT-B/32 as the default backbone for ours and baseline fine-tuning methods in our manuscript and also evaluate CLIP ViT-B/16 and CyCLIP ResNet50 in Section C of this Supplementary Material.

**Baseline Methods**  As standard baselines, we first consider zero-shot CLIP (ZS) and vanilla fine-tuning (FT) with contrastive loss. Then, we perform exhaustive ablation ($V$-, $L$-, $VL$-, and $m^2$-Mix) as well as our complete objective $m^3$-Mix with its high-temperature variant. To further compare our approach with state-of-the-art fine-tuning methods, we consider MaPLe [64] that optimizes the continuous prompts inside the text and image encoders of CLIP, and the contrastive loss extended version of uni-modal fine-tune methods: LP-FT [11] which trains classification head and full modal separately in a two-stage manner, and WiSE-FT [10] which performs parameter-space ensemble between the pre-trained checkpoint and fine-tuned checkpoint. Additionally, we consider the ES and our $m^3$-Mix as the *plug-in* methods to improve the above three state-of-the-art fine-tuning methods that are denoted as method names w/ ES or $m^3$-Mix in Tab. 4 and 5 of the main paper.

**Metric**  For both the few-shot adaptation and distribution shift setting, we report top-1 accuracy as the In-Distribution Accuracy (ID Acc.) and Out-Of-Distribution Accuracy (OOD Acc.), respectively.

**Implementation Detail**  In this paper, we propose new contrastive losses $m^2$-Mix and $m^3$-Mix which consume the image-text paired instances. However, the above datasets provide class name labels only and do not have captions corresponding to each image. To make CL methods amendable for this setting, we adopt a common prompt 'a photo of classname' that wraps the class name with a short context and use this as captions of images. Different from image-caption-based contrastive learning on Flickr30k and MS COCO, a batch of ImageNet-1K contains multiple samples that are assigned to the same class. We construct the label map for contrastive loss by regarding all of the samples from a class as positives. Following [82, 64], we perform the tasks under the same few-shot evaluation protocol: 16-shot training samples per class and inference on the entire test set. To construct the contrastive loss, we first compute the pivot classifier embedding by forwarding all possible class category names to the text encoder. Then, we calculate the pairwise similarity between in-batch image embedding and pivot embedding and construct the label matrix by reflecting the fact that there are many positive images corresponding to a text embedding (for each class). To implement the contrastive loss with multi-modal Mixup, we mix the in-batch image embedding and text embedding and contrast the resulting mixed embedding with image and pivot embedding, respectively.

About training configuration, in the distribution shift setting, we train all methods (except MaPLe) on 20 epochs with batchsize 100 via AdamW optimizer with default parameters. Due to MaPLe's huge memory requirements, we set the batchsize to 4 and train a single epoch. As shared hyperparameters, we pivot the initial learning rate to 1e-6 and search for the best maximum learning rate from {1e-6, 3e-6, 5e-6, 7e-6, 1e-5} and weight decay from {0, 1e-3, 5e-3, 1e-2, 5e-2 1e-1} for all training methods (except MaPLe's learning rate sweep from {5e-3, 1e-3, 5e-4, 1e-4}). Here we use the one-cycle

---

[9]We scheduled the strength of sum of the Mixup-based loss terms by `L_mix/epoch`

cosine learning rate scheduler. For the few-shot adaptation in a general setting, we train each method over 200 epochs (40 epochs for MaPLe) with the same batchsize, optimizer, and hyperparameter sweep range. In both two settings, we use the same data augmentation procedure (random resize crop and random flip) [82, 64] for all methods. Note that for LP-FT, we train both the linear head and full models during half of the entire epochs, and we do not use data augmentation in the linear head training phase following the authors' proposal. For our methods, weighting coefficient of $m^2$-Mix and uni-modal mixups are explored over {0.01, 0.1, 0.2, 0.3, 0.4, 0.5}, and the parameters of Beta distribution are swept over {0.2, 0.5}.

### A.3 Multi-modal Classification

**Dataset** To evaluate the multi-modal representation learning under video emotional classification, we consider the CMU-MOSEI [92], a popular benchmark for multi-modal sentiment analysis. CMU-MOSEI consists of three modalities textual (T), visual (V), and audio (A), and contains 23,453 YouTube video clips about diverse movie reviews, and each clip is annotated with ordinal labels ranging from -3 (strong negative) to 3 (strong positive). In the training phase, three modalities are fully available to all methods, and only one or two modalities are given in the evaluation phase to measure the robustness under modality missing as well as the informativeness of individual-modality representations.

**Model Description and Baseline Methods** To construct backbone architecture, following Poklukar et al. [84], we adopt the Multimodal Transformer (MulT) [86] as a joint-modal encoder which enables the commutation among modalities with cross-modal attention block. To enhance the explicit alignment between modalities, Poklukar et al. [84] propose Geometric Multimodal Contrastive Learning (GMC). In addition to the joint encoder. GMC introduces lightweight modality-specific encoders constructed by a single Gated Recurrent Unit [93] followed by a linear projection layer, then performs contrastive learning between joint representation (from a joint encoder) and uni-modal representation (from modality-specific encoders). We set the MulT as a standard baseline, GMC as a contrastive learning-enhanced baseline, and then plug our $m^2$-Mix to GMC objective to validate whether our method can give additional benefits to multi-modal representation learning.

While MulT learns the joint encoder only with standard classification loss (i.e., cross-entropy loss; $L_{ce}$), GMC learns joint and modality-specific encoders with the objective function $L_{ce} + L_{GMC}$ where $L_{GMC}$ deals with the sum of all one-to-joint contrastive losses. On top of GMC, $L_{m^2\text{-Mix}}$ is integrated with a trade-off hyperparameter $\beta$: $L_{ce} + L_{GMC} + \beta L_{m^2\text{-Mix}}$.

**Metric** As mentioned earlier, in the inference time, each method can encounter partial modalities among T, V, and A. To this end, we evaluate each method under 7 environments sorted by the available modalities: (T), (V), (A), (T,V), (T,A), (V,A), (T,V,A). Then, we measure the classification accuracy, F1-score (in supplementary material), uniformity, and alignment.

**Implementation Detail** We train all methods over 40 epochs with batchsize 128 via Adam optimizer with the default configuration. Following [84], we set the learning rate to 1e-3 and do not apply the weight decay. The trade-off parameter $\beta$ and the parameter $\alpha$ of Beta distribution Beta($\alpha$, $\alpha$) are optimized among {0.1, 0.2, 0.3, 0.4, 0.5} and {0.5, 1.0, 1.5, 2.0}, respectively. Those are selected $\beta = 0.2$ and $\alpha = 2.0$. To implement the contrastive loss with $m^2$-Mix, we randomly sample two modalities for each training epoch and mix them to build a mixed representation. Then, we compute (1) the positive similarity between paired mixed and joint representation and (2) the negative similarity between non-paired mixed and joint representation. Finally, we compute the modified $L_{m^2\text{-Mix}}$ as a negative logarithm of the sum of positive similarities over the sum of negative similarities. Different from the CLIP fine-tuning cases, we use the multi-modal mixed representation as both positive and negative pairs with the target joint representation because our goal in this task is to align the embedding between the joint and other modalities. For inference with partial modalities, we average the representations from given modalities to make a single embedding that is sent to the classifier head for a fair comparison with [86, 84].

### A.4 Multi-modal Mixup for Contrastive Captioner

**Dataset**    To demonstrate the effectiveness of the multi-modal Mixup on a state-of-the-art vision-language model, Contrastive Captioner (CoCa), we perform cross-modal retrieval on Flickr30k and MS COCO, and image captioning task on MS COCO (that of 2014).

**Model Description and Baseline Methods**    We adopt LAION-2B pre-trained CoCa ViT-L/14 from `OpenCLIP` library as our backbone model, and consider three learning objectives for CoCa fine-tuning: (1) autoregressive captioning loss (Cap), (2) contrastive loss and captioning loss (CL + Cap), and (3) contrastive loss, $\mathcal{L}_{m^2\text{-Mix}}$, and captioning loss (CL w/ $\mathcal{L}_{m^2\text{-Mix}}$ + Cap).

**Metric**    For image-text retrieval, we adopt top-1 and top-5 recalls likewise CLIP retrieval setup. For image captioning, five standard metrics: BLEU-4 [94], METEOR [95], ROUGE-L [96], CIDEr [97], and SPICE [98] are evaluated.

**Implementation Detail**    For all three methods, we train the model on MS COCO over one epoch with `OpenCLIP`[10]-provided hyperparameter configuration, i.e., 128 as batch size, 1e-5 as learning rate, 0.1 as weight decay, and 1000 as learning rate warm-up steps. After fine-tuning on MS COCO, we evaluate the model on MS COCO for fine-tuned image-text retrieval and image captioning, and on Flickr30k for zero-shot transferred image-text retrieval. Here, we adopt `CLIP_benchmark`[11] library for easy evaluation. For training of CL + Cap, we weight $\mathcal{L}_{CL}$ as 1.0 and $\mathcal{L}_{Cap}$ as 2.0. Our $m^2$-Mix related parameters are explored over {0.1, 0.2, 0.3, 0.4, 0.5, 1.0} for Beta distribution parameter and {0.1, 0.2, 0.25, 0.3, 0.35, 0.4, 0.5, 0.7, 1.0} for $\mathcal{L}_{m^2\text{-Mix}}$ weighting coefficient.

---

[10]https://github.com/mlfoundations/open_clip
[11]https://github.com/LAION-AI/CLIP_benchmark

# B Pseudo Code

**Algorithm 1:** PyTorch-style Implementation Code for Geodesic Multi-Modal Mixup

```python
# X,Y : image batch, text batch
# f,g :  learnable image encoder and text encoder
# t1, t2 :  trainable temperature parameters
# alpha1, alpha2 :  parameters for Beta Distribution
# args.{m2mix, vmix, lmix, vlmix} :  weighting parameters
def ce(logits,targets):
   return (-targets*nn.LogSoftmax(dim=-1)(logits)).sum()
def cross_entropy_2D(logits,targets):
   return ((ce(logits,targets)+ce(logits.T,targets.T))/2).mean()
def geodesic_mix(lambda,a,b):
   theta = torch.acos( (a*b).sum(dim=[1])).view(a.shape[0],1)
   n1 = torch.sin(lambda*theta)/torch.sin(theta)*a
   n2 = torch.sin((1-lambda)*theta)/torch.sin(theta)*b
   return n1+n2

def ContrastiveLoss(X,Y,f,g,t1,t2,args)
   I = torch.eye(X.shape[0])
   I_R = torch.flip(I,dims=[0])
   I_X, I_XD = I+I_R, 1-(I+I_R)
   If, Tf = f(X), g(Y) # L2 normalized features
   logits = If@Tf.T # Original logit
   loss = cross_entropy_2D(logits/t1,I) # 2D Cross Entropy
   if args.m2mix:
      lambda = random.betavariate(alpha2,alpha2)
      mix = geodesic_mix(lambda,If,Tf)
      logits2_i = mix@Tf.T
      logits2_i = logits*I + logits2_i*(1-I)
      logits2_t = mix@If.T
      logits2_t = logits.T*I + logits2_t*(1-I)
      loss += args.m2mix*(ce(logits2_i/t2,I) + ce(logits2_t/t2,I))/2
   if args.vmix:
      lambda = random.betavariate(alpha1,alpha1)
      mix = geodesic_mix(lambda,If,If.flip())
      logits2 = mix@Tf.T
      logits2 = logits2*I_X + logits*I_XD
      loss += args.vmix*cross_entropy_2D(logits2/t1,lambda*I +
        (1-lambda)*I_R)
   # L-Mix is omitted
   if args.vlmix:
      lambda = random.betavariate(alpha1,alpha1)
      mix_I = geodesic_mix(lambda,If,If.flip())
      mix_T = geodesic_mix(lambda,Tf,Tf.flip())
      logits2 = mix_I@mix_T.T
      logits2 = logits2*I + logits*(1-I)
      loss += args.vlmix*cross_entropy_2D(logits2/t1,I)
   return loss
```

# C  Additional Results

## C.1  Mixed Embedding Analysis

In Fig. 8, we post examples of image-retrieval results by our $m^2$-Mix. Big images and text on the left denote the original image and text pair. The right top and bottom denote the top-3 retrieved images from the original image embedding and mixed embedding, respectively. Overall, retrieved images by $m^2$-Mixed embedding contain more rich semantics that is derived from both image and text.

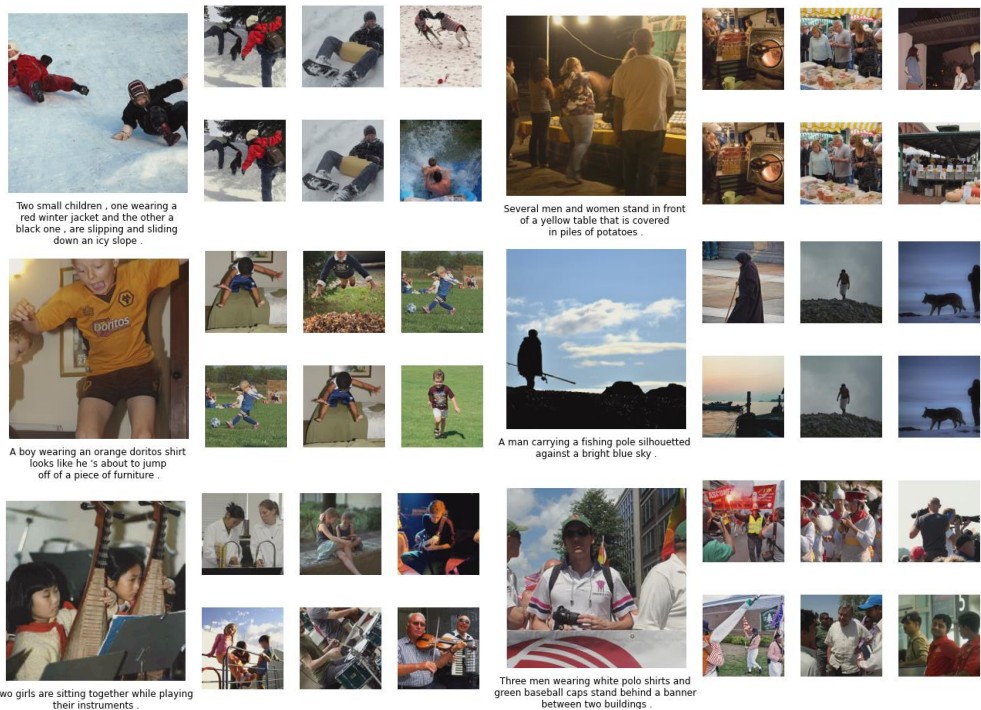

Figure 8: Retrieved images by original image embedding and mixed embedding on Flickr30k.

## C.2  Additional Results on Uni-modal Classification

This section provides results of 16-shot uni-modal classification on four new datasets (EuroSAT [99], FGVC Aircraft [100], UCF101 [101], Stanford Cars [102]) with two different models (CLIP ViT-B/16 and CyCLIP [33] ResNet50) that are lacking in the main paper. We perform fine-tuning of them from their official checkpoint relase[12][13] with the same hyperparameter sweep range described in A of Supplementary. In Table 11, our $m^2$-Mix brings consistent performance gain across all datasets and models with some significant boosting on the FGVC Aircraft and Stanford Cars datasets. Thus, $m^2$-Mix is a general approach that can enhance the representation learning on various settings.

Next, in Figure 9 and 10, we perform ablation on the parameter $\alpha$ of Beta distribution, which stochastically determines the mixing ratio between two modalities. Red and Blue colors denote a constant parameter and a linear scheduling parameter, respectively. We see that lower value $\alpha$ (U-shaped Beta distribution) generally achieves better performance than larger values (uniform or reversed U-shaped) on the two classification datasets.

Linear scheduling of Beta parameters drives promising results in some cases, e.g., $1.0->0.1$ and $2.0->0.1$ in Stanford Cars. It seems crucial to enforce that the shape of Beta distribution ends up with a U-shape for the success of scheduling variants. That is, the small-to-many mixing fashion is better than that of half-to-half for the geodesic multi-modal Mixup on classification.

---

[12]https://github.com/openai/CLIP
[13]https://github.com/goel-shashank/CyCLIP

Table 11: Few-shot classification results with CLIP-VIT-B/16 and CyCLIP-RN50 models. We follow the same few-shot evaluation protocol and contrastive learning strategy with Sec 5.4. in our manuscript. $m^2$-Mix consistently outperforms the baseline methods across four datasets. Especially, on Aircraft, $m^2$-Mix achieves 8.5% and 4.7% gain over FT in CLIP and CyCLIP, respectively, and 5.2% gain over FT in CyCLIP on Cars.

| Model | Method | Dataset | | | |
|---|---|---|---|---|---|
| | | EuroSAT | Aircraft | UCF101 | Cars |
| CLIP (ViT-B/16) | ZS | 48.41 | 24.81 | 67.46 | 65.33 |
| | FT | 94.03 | 60.61 | 86.36 | 88.58 |
| | FT w/ $m^2$-Mix | **94.33** | **69.07** | **86.94** | **90.36** |
| CyCLIP (RN50) | FT | 84.98 | 48.19 | 67.25 | 67.02 |
| | FT w/ $m^2$-Mix | **85.22** | **52.96** | **68.97** | **72.22** |

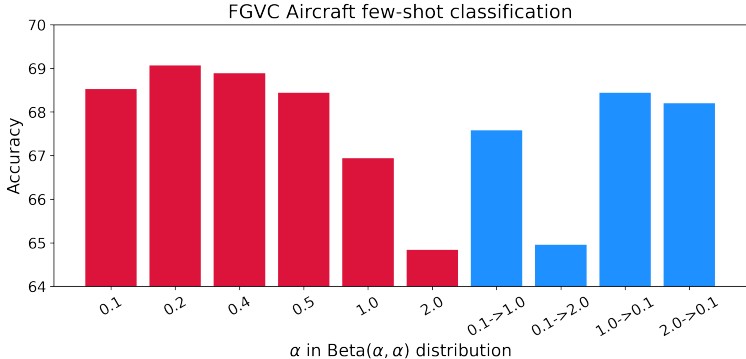

Figure 9: Few-shot classification results on FGVC Aircraft dataset. We varied the parameter of Beta distribution, constant (red) or scheduled (blue), to simulate diverse situations of mixed samples. It always achieves better performance than FT (60.61), even under varying parameters.

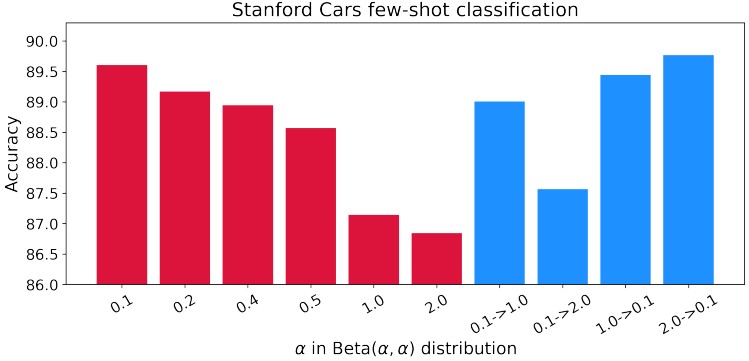

Figure 10: Few-shot classification results on Stanford Cars dataset. We varied the parameter of Beta distribution, constant (red) or scheduled (blue), to simulate diverse situations of mixed samples. It generally achieves better performance than FT (88.58), even under varying parameters.

## C.3 Additional Results on Multi-modal Classification

In addition to classification accuracy (in the main paper), we additionally present the F1-score (in Tab. 12) for diagnosis on classification results. While GMC with $m^2$-Mix is outperformed by GMC in two of three single-modality cases, it shows superior results on two-modality-given cases based on explicit enforcement of bi-to-joint alignment during training.

Table 12: Classification F1-score on CMU-MOSEI under complete and partial observation modalities. We report the mean performance and standard deviation of five runs.

| Method | Test-time Observed Modalities | | | | | | |
|---|---|---|---|---|---|---|---|
| | Full(T+V+A) | T | V | A | T+V | T+A | V+A |
| MulT | 0.8056±0.004 | 0.6909±0.051 | 0.5678±0.107 | 0.6021±0.151 | 0.6453±0.096 | 0.6657±0.097 | 0.5922±0.111 |
| GMC | 0.8054±0.001 | 0.7846±0.006 | **0.6548±0.008** | **0.6910±0.008** | 0.7747±0.009 | 0.7810±0.003 | 0.6978±0.004 |
| GMC+$m^2$-Mix | **0.8086±0.001** | **0.7882±0.005** | 0.6522±0.006 | 0.6875±0.0080 | **0.7814±0.004** | **0.7840±0.004** | **0.6988±0.003** |

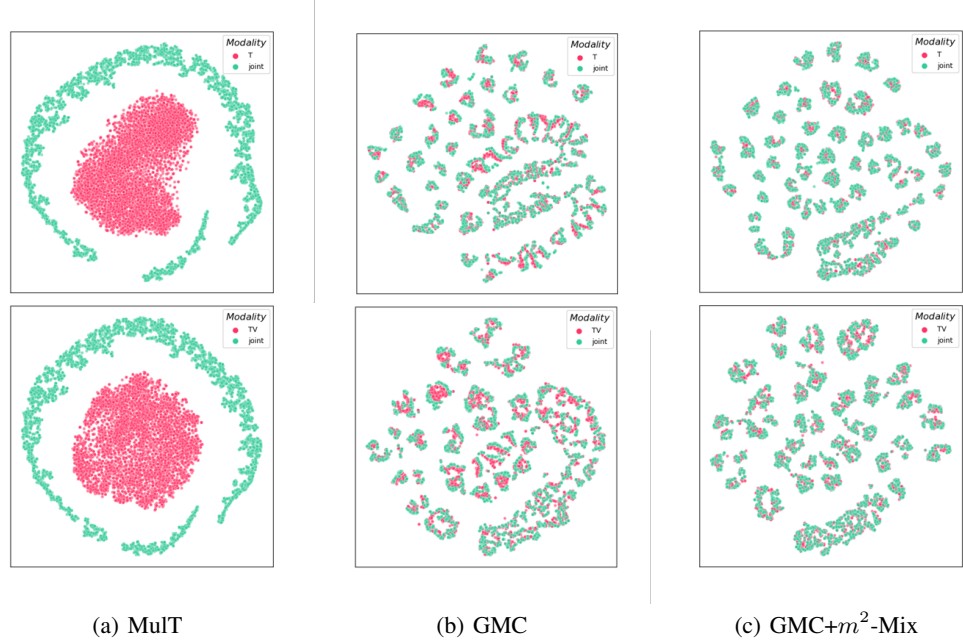

(a) MulT        (b) GMC        (c) GMC+$m^2$-Mix

Figure 11: t-SNE [85] for CMU-MOSEI, which has textual (T), visual (V), and audio (A) modalities. Top row represents when the only textual (T) information is given, and the bottom row corresponds to when the textual (T) and visual (V) information are given. The pink and green color denotes the embedding of partial and joint modality.

Fig. 11 shows the embedding t-SNE of each method given one (top row) or two (bottom row) modalities in test-time. Compared with MulT, GMC strongly aligns the embedding of partial and joint modality based on its explicit enforcement, and the alignment is further enhanced by the aid of $m^2$-Mix, which results in superior performance (in Tab. 7 of the main paper, Tab. 12 of supplementary) when only partial (missing modality) information is given during test-time. These results justify the use of our $m^2$-Mix for robust multi-modal representation learning under missing modality scenarios.

## C.4 Effect of $m^2$-Mix on Contrastive Learning

This section provides a more detailed analysis of $m^2$-Mix. Specifically, we present (i) the proportion of negative pairs (original and mixed) that exceed the similarity between that of positive pairs (See Fig. 12 - 15), and (ii) the similarity comparison between positive and original negative pairs with and without $L_{m^2\text{-Mix}}$ (See Fig. 16 and 17). All results are from cross-modal retrieval with CLIP ViT-B/32 on Flickr30k and MS COCO.

Fig. 12 - 15 show the average proportion of in-batch negative pairs' similarities that exceeds the similarities of positive pairs during training iterations (dataset: MS COCO - Fig. 12 and 13, Flickr30k - Fig. 14 and 15, similarity computation: $I$-to-mixed - Fig. 12 and 14, $T$-to-mixed - Fig. 13 and 15).

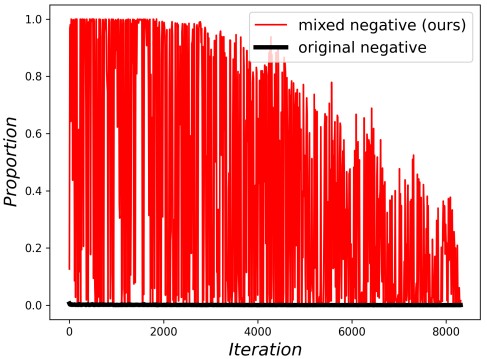

Figure 12: Hard negative proportion by $I$-to-mixed samples' similarities on MS COCO.

Figure 13: Hard negative proportion by $T$-to-mixed samples' similarities on MS COCO.

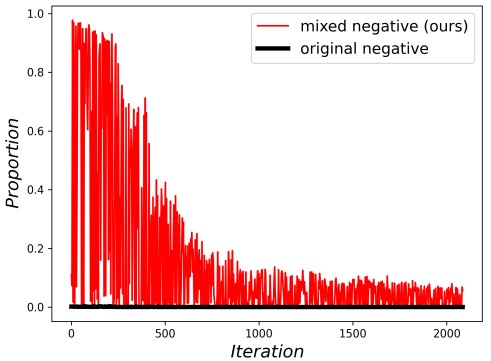
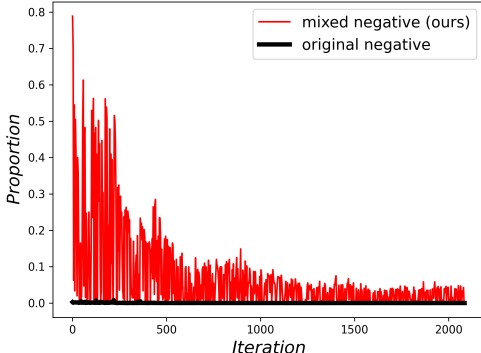

Figure 14: Hard negative proportion by $I$-to-mixed samples' similarities on Flickr30k.

Figure 15: Hard negative proportion by $T$-to-mixed samples' similarities on Flickr30k.

When we train our model with learning objective $L_{\text{CLIP}} + L_{m^2\text{-Mix}}$, the similarities of negative pairs by our $m^2$-Mix significantly higher than not only original negatives' similarities but also positive similarities. Especially in the early training iterations, the proportion is about one in Fig 12 - 14, i.e., almost all of the in-batch negative pairs have higher similarities than positive ones. These results advocate our assumption in proof of Proposition 4.2. (of the main paper and Proposition 1 of this supplementary). That is, $m^2$-Mix-generated negative pairs empirically have higher similarities than positive ones. Even though such a hard negative proportion is decaying as the training progresses (by pursuing alignment), it still has not vanished to zero, i.e., uniformity is encouraged until the end of training while the magnitude is getting weaker.

Next, in Fig. 16 left and 17 left, we present the in-batch averaged pairwise similarity from normal negative pairs and positive pairs, which contribute to the computation of $L_{\text{CLIP}}$. We evaluate the similarities under two scenarios whether our $L_{m^2\text{-Mix}}$ is adopted together with $L_{\text{CLIP}}$ (dashed line) or not (solid line). Among negative pair similarities, we only consider that of the top-1 hardest negative pair that is related to the definition of relative alignment (Eq. 2 in the main paper) and contrastive loss in an asymptotic scheme (Theorem 4.2. in the main paper).

As we can see in both Fig. 16 and 17, the top-1 **original negatives'** similarities and positives' similarities are increased by using $m^2$-Mix, even though the hard negatives generated by $m^2$-Mix are not explicitly consumed in $L_{\text{CLIP}}$. Consequently, the alignment is strongly enhanced (left side of Fig. 16 and 17). The results can be summarized as follows: (1) Whether $L_{m^2\text{-Mix}}$ is used or not, the similarities between positive pairs are larger than that of original negative pairs during the whole training time. (2) However, if $L_{m^2\text{-Mix}}$ is used with $L_{\text{CLIP}}$, the top-1 negative similarities are increased. (3) As a result, the similarities between positive pairs are increased to decrease the contrastive loss $L_{\text{CLIP}}$ (which converges asymptotically to triplet loss).

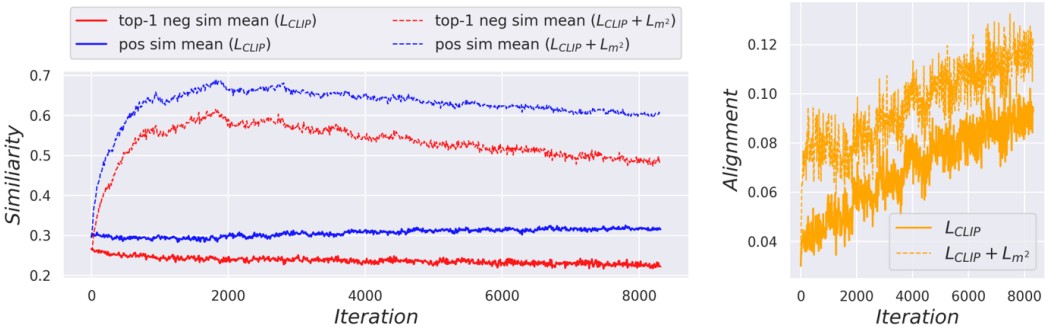

Figure 16: In-batch averaged pairwise similarity and alignment comparison over training iterations with and without $m^2$-Mix on MS COCO. (Left) similarities of original negative pairs (top-1 highest) and that of positive pairs. (Right) alignment comparison.

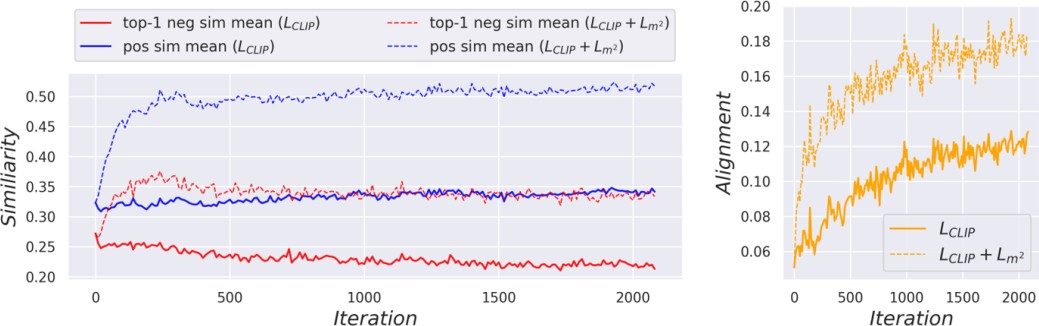

Figure 17: In-batch averaged pairwise similarity and alignment comparison over training iterations with and without $m^2$-Mix on Flickr30k. (Left) similarities of original negative pairs (top-1 highest) and that of positive pairs. (Right) alignment comparison.

These empirical results imply that our $L_{m^2\text{-Mix}}$ implicitly helps the alignment in $L_{\text{CLIP}}$, and by minimizing $L_{\text{CLIP}} + L_{m^2\text{-Mix}}$, we can deal with both alignment and uniformity better (Proposition 4.2. of main paper).

# D Proof

**Theorem D.1** (Hardness of $m^2$-Mixed samples). *Let's assume that two random variables $x_1$ and $x_2$ follow the $M_d(\mu_1, \kappa)$ and $M_d(\mu_2, \kappa)$, von Mises–Fisher distribution with mean direction $\mu_1, \mu_2$ and concentration parameter $\kappa$ in $\mathbb{R}^d$, respectively. Let $\widetilde{x} = x_1 + x_2$ and $d = 2$. Then, $D_{KL}(p(x_1)||p(\widetilde{x})) \leq D_{KL}(p(x_1)||p(x_2))$ for sufficiently large $\kappa$.*

*Proof of Theorem D.1.* Let $M_d(\mu, \kappa) = C_d(\kappa)\exp(\kappa\mu^T x)$, where $C_d(\kappa) = \frac{\kappa^{d/2-1}}{(2\pi)^{(d/2)}I_{d/2-1}(\kappa)}$. Let $x_1 \sim M_2(\mu_1, \kappa)$ and $x_2 \sim M_2(\mu_1, \kappa)$, and $I$ denotes the modified Bessel function. From [103], $\widetilde{x} \sim M_2(\widetilde{\mu}, \widetilde{\kappa})$ where $\widetilde{\mu} = \mu_1 + \mu_2$ and $\widetilde{\kappa} = A^{-1}(A(\kappa)A(\kappa))$, approximately, where $A(\kappa) = \frac{I_1(\kappa)}{I_0(\kappa)}$ and $A^{-1}$ is its inverse. From [104], $D_{KL}(p(x_1)||p(x_2)) = \kappa A(\kappa)(1 - \cos(\mu_1 - \mu_2))$. Similarly, $D_{KL}(p(x_1)||p(\widetilde{x})) = \log I_0(\widetilde{\kappa}) - \log I_0(\kappa) + \widetilde{\kappa}A(\widetilde{\kappa})(1 - \cos(\mu_1 - \mu_2))$. From [105] and [106], $I_0(\kappa) \approx \frac{\exp(\kappa)}{\sqrt{2\pi\kappa}}(1 + \frac{1}{8\kappa})$ and $I_1(\kappa) \approx \frac{\exp(\kappa)}{\sqrt{2\pi\kappa}}(1 - \frac{3}{8\kappa})$ for sufficiently large $\kappa$. Therefore, $D_{KL}(p(x_1)||p(\widetilde{x})) \leq D_{KL}(p(x_1)||p(x_2))$ for sufficiently large $\kappa$. $\square$

**Proposition D.2** (Limiting behavior of $\mathcal{L}_{\text{CLIP}}$ with $m^2$-Mix). *For sufficiently large $M$, as the temperature of contrastive loss $\tau \to 0^+$, the $\mathcal{L}_{CLIP}$ and $\mathcal{L}_{m^2\text{-Mix}}$ converge to the triplet loss with zero-margin (i.e., corresponding to negative Alignment) and negative Uniformity, respectively. That is:* $\lim_{\tau \to 0^+} \mathcal{L}_{CLIP} + \mathcal{L}_{m^2\text{-Mix}} \simeq -(Alignment + Uniformity)$

*Proof of Proposition D.2.* As noted in Section 3., given a training batch $\{x_i, y_i\}_{i=1}^M$ of image-text pairs and image and text encoders $f(\cdot; \theta_1)$ and $g(\cdot; \theta_2)$, and the $L_2$-normalized image-text embeddings are $(I, T) = \{f(x_i; \theta_1), g(y_i; \theta_2)\}_{i=1}^M$. Then, for $\theta = \{\theta_1, \theta_2\}$ and a scalar $\tau > 0$, a standard contrastive loss adopted by CLIP is formulated as:

$$C(I, T; \theta) = \frac{1}{M}\sum_{i=1}^M -\log\frac{\exp((I_i \cdot T_i)/\tau)}{\sum_{j=1}^M \exp((I_i \cdot T_j)/\tau)} \tag{6}$$

Although the actual loss function is constructed by averaging two-way contrastive losses, i.e., $L_{CL} = \frac{1}{2}(C(I, T; \theta) + C(T, I; \theta))$, here, we consider only one-way contrastive loss $C(I, T; \theta)$ for simplicity. It is easily shown the derivation of the other side by changing the order of $I$ and $T$.

During pre-training, the temperature $\tau$ of CLIP converges to 0.01, which is a significantly small value that makes the pair-wise similarities sharp. Thus, it will be reasonable to consider an extreme case: when $\tau \to 0^+$. In this case, we can approximate the $C(I, T; \theta)$ as follow:

$$
\begin{aligned}
C(I, T; \theta) &= \lim_{\tau \to 0^+} \frac{1}{M}\sum_{i=1}^M -\log\frac{\exp(I_i \cdot T_i/\tau)}{\sum_{j=1}^M \exp(I_i \cdot T_j/\tau)} \quad (7)\\
&= \lim_{\tau \to 0^+} \frac{1}{M}\sum_{i=1}^M -(I_i \cdot T_i)/\tau + \log\left[\exp(I_i \cdot T_i/\tau) + \sum_{j \neq i}\exp(I_i \cdot T_j/\tau)\right]\\
&= \lim_{\tau \to 0^+} \frac{1}{M}\sum_{i=1}^M \log\left[1 + \sum_{j \neq i}\exp((I_i \cdot T_j) - (I_i \cdot T_i)/\tau)\right]\\
&= \lim_{\tau \to 0^+} \frac{1}{M}\sum_{i=1}^M \log\left[1 + \sum_{j \in \mathcal{J}(i, I, T)}\exp((I_i \cdot T_j) - (I_i \cdot T_i)/\tau)\right]\\
&\qquad\qquad (\text{where } \mathcal{J}(i, I, T) := \{j | (I_i \cdot T_j) > (I_i \cdot T_i)\})\\
&= \lim_{\tau \to 0^+} \frac{1}{M}\sum_{i=1}^M \frac{1}{\tau}\max\left[\max_j(I_i \cdot T_j) - (I_i \cdot T_i), 0\right]\\
&\simeq \lim_{\tau \to 0^+} -\text{Alignment}(I, T; \theta)\\
&\qquad (\text{when } \max_j(I_i \cdot T_j) - (I_i \cdot T_i) > 0 \text{ and } M \text{ is sufficiently large})
\end{aligned}
$$

where $\max_j(I_i \cdot T_j)$ denotes the maximum similarity among negative pairs. From this derivation, we show that the multi-modal contrastive loss only considers the top-1 hardest negative pairs to positive ones like a triplet loss. As a result, minimizing this loss function is equivalent to maximizing the relative alignment that we newly defined in this paper (in Section 3. Eq. 2 of main paper), e.g., for sufficiently large $M$, $\underset{\theta}{\text{minimize}}\ C(I, T; \theta) \equiv \underset{\theta}{\text{maximize}}\ \text{Alignment}(I, T; \theta)$. Note that, however, if there are no hard negatives that have higher similarity than positives, the above loss term does not give a meaningful learning signal because the loss already approaches zero. This issue is resolved by $m^2$-Mix, which generates hard negatives explicitly.

Next, when we consider the $m^2$-Mix-based contrastive loss under same case ($\tau \to 0^+$), we can obtain another approximation like below:

$$
\begin{aligned}
C_{m^2\text{-Mix}}(I, T; \theta) &= \lim_{\tau \to 0^+} \frac{1}{M} \sum_{i=1}^{M} - \log \frac{\exp(I_i \cdot T_i / \tau)}{\exp(I_i \cdot T_i / \tau) + \sum_{j \neq i} \exp(I_i \cdot m_\lambda(I_i, T_j) / \tau)} \quad (8) \\
&= \lim_{\tau \to 0^+} \frac{1}{M} \sum_{i=1}^{M} -(I_i \cdot T_i)/\tau + \log \left[ \exp(I_i \cdot T_i / \tau) + \sum_{j \neq i} \exp(I_i \cdot m_\lambda(I_i, T_j) / \tau) \right] \\
&= \lim_{\tau \to 0^+} \frac{1}{M} \sum_{i=1}^{M} \log \left[ 1 + \sum_{j \neq i} \exp((I_i \cdot m_\lambda(I_i, T_j) - (I_i \cdot T_i))/\tau) \right] \\
&= \lim_{\tau \to 0^+} \frac{1}{M} \sum_{i=1}^{M} \log \left[ 1 + \sum_{j \neq i} \exp(I_i \cdot m_\lambda(I_i, T_j) / \tau) \right] \\
&\qquad\qquad\qquad\qquad \text{(by assuming } I_i \cdot m_\lambda(I_i, T_j) > I_i \cdot T_i \text{ for all } j \neq i) \\
&= \lim_{\tau \to 0^+} \frac{1}{M} \sum_{i=1}^{M} \log \sum_{j \neq i} \exp(I_i \cdot m_\lambda(I_i, T_j) / \tau) \\
&\simeq \lim_{\tau \to 0^+} -\text{Uniformity}(I, m_\lambda(I_i, T_j); \theta) \qquad\qquad \text{(for sufficiently large } M)
\end{aligned}
$$

Where $m_\lambda(\cdot, \cdot)$ is the geodesic Mixup operation with mixing ratio $\lambda$. Based on above Eq. 8, we argue that our $m^2$-Mix asymptotically maximizes the uniformity given sufficiently in-batch large samples $M$, i.e., $\underset{\theta}{\text{minimize}}\ C_{m^2\text{-Mix}}(I, T; \theta) \equiv \underset{\theta}{\text{maximize}}\ \text{Uniformity}(I, m_\lambda(I_i, T_j); \theta)$.

To complete the proof, consider the two-side contrastive losses $L_{\text{CLIP}} = \frac{1}{2}(C(I, T; \theta) + C(T, I; \theta))$ and $L_{m^2\text{-Mix}} = \frac{1}{2}(C_{m^2\text{-Mix}}(I, T; \theta) + C_{m^2\text{-Mix}}(T, I; \theta))$. Then, by minimizing the combination of the standard contrastive loss $L_{\text{CLIP}}(\theta)$ (Eq. 7) and $L_{m^2\text{-Mix}}(\theta)$ (Eq. 8), we can approximately maximize both Alignment and Uniformity. $\qquad\qquad\square$

## E  Limitation

In this work, we observed that uniformity and alignment (our modified formulation) are both somewhat correlated with downstream performance, so we argued that uniformity and alignment are crucial in multi-modal representation learning as well as uni-modal representation learning [25, 26]. However, those two metrics are not absolute ones for various situations, e.g., when the modality-specific unique information is important, the higher uniformity and alignment can cause the loss of modality-specific information. The increased computational cost for additional contrastive loss terms is another limitation.