# OpenReview forum: "Geodesic Multi-Modal Mixup for Robust Fine-Tuning"
_NeurIPS.cc/2023/Conference — NeurIPS 2023 poster_

### Official Review · Reviewer_iPuA · 2023-06-09

**Soundness:** 4 excellent
**Presentation:** 4 excellent
**Contribution:** 4 excellent
**Rating:** 7
**Confidence:** 5

**Summary:**

The paper proposes Geodesic Multi-Modal Mixup, which mixes heterogeneous modality embeddings on the hypersphere to generate harder negative samples for the contrastive loss of multi-modal models. Such training scheme improves modality alignment and uniformity. The method was evaluated on various tasks including retrieval, calibration, few-shot classification, and embedding arithmetic.

**Strengths:**

- The multi-modal mixup approach is proposed opportunely.
- The method is comprehensive and straightforward.
- The idea of mixing different modalities is novel and interesting.
- Extensive experiments are impressive with significant gains in several settings.

**Weaknesses:**

1. I believe the key hyperparameter of the proposed method is lambda, as this should control the "hardness" of the mixed samples.  Please provide an analyses (at least a sensitivity test) on this component.
2. In some cases, the authors only report m^2-Mix, while others report only m^3-Mix. It would be great if the authors could provide full tables with both mixup methods, and discuss reasons if there are some differences in performance between the two.
3. Minor presentation issues:
- Some figures have too small a font size; e.g. Figure 1,6.
- Eq. (5) and L202-203 are confusing at first glance. Please use separate lines or a comma in between.
- Figure 6 right's title should be "GMC w/ M^2-Mix"?


**Questions:**

Overall, the paper was very well-written with plenty of applications. The mentioned weaknesses are relatively minor, and no critical issue was found on my side.

**Limitations:**

Yes

---

> ### Author Rebuttal · Authors · 2023-08-10
>
> Thank you for your constructive comments and suggestions, and they are exceedingly helpful for us to improve our paper. We are grateful that you find strength in our idea, actual method, and experiments!
>
> > **Comment1)** I believe the key hyperparameter of the proposed method is lambda, as this should control the "hardness" of the mixed samples. Please provide analyses (at least a sensitivity test) on this component.
>
> To address your suggestion, we present sensitivity analyses (Fig. 1 and Fig. 2 in the global response) on parameters of Beta distribution that contribute to the mixing rate between two embeddings. From the experiments, we would like to remark follows:
> * $m^2$-mix generally shows stronger performances when the parameter $\alpha$ lay between 0.1 and 0.5 (U-shaped) rather than larger values (uniform or reversed U-shaped). This is consistent with the well-worked interval that was proposed in the original Mixup [Zhang et al. 2018] paper but different from the selected value by Manifold Mixup (alpha=2.0) [Verma et al. 2019], and implies that the many-to-small strategy is better than the half-to-half mixing strategy in our geodesic multi-modal mixup.
> * The linear scheduling variants of Beta parameters (0.1 -> 1.0, 0.1 -> 2.0, and their opposite direction counterparts) achieve promising results in some cases, e.g., 1.0 -> 0.1 and 2.0 -> 0.1 in Stanford Cars. For scheduling variants, it is important to ensure that the shape of Beta distribution approaches to U-shape at the endpoints of learning.
> * While scheduling-based variants have shown promising results in some cases, they are not significantly better than the fixed-parameter strategy, so for the sake of simplicity, we maintain the fixed-parameter strategy for the multi-modal mixup.
>
>
> > **Comment2)** In some cases, the authors only report $m^2$-Mix, while others report only $m^3$-Mix. It would be great if the authors could provide full tables with both mixup methods, and discuss reasons if there are some differences in performance between the two.
>
> While we conducted the experiments (in the global response) only with $m^2$-Mix for this rebuttal due to limited time and resources, we agree with your suggestion and have planned to put both $m^2$-Mix and $m^3$-Mix in all tables consistently on the final version of the paper and provide detailed discussions on differences in performance.
>
>
> > **Comment3)** Minor presentation issues
>
> Thank you for taking the time to review the details as well as primary messages in our paper, we will reflect on all your remarks to revise our paper.
>
>
> ---
>
> [Zhang et al. 2018] Zhang, Hongyi, et al. "mixup: Beyond Empirical Risk Minimization." International Conference on Learning Representations. 2018.
> [Verma et al. 2019] Verma, Vikas, et al. "Manifold mixup: Better representations by interpolating hidden states." International conference on machine learning. PMLR, 2019.

---

> > ### Comment · Reviewer_iPuA · 2023-08-16
> > **Rebuttal Acknowledgement**
> >
> > Thank you for your detailed responses. Most of my questions were answered. I'll keep my score.

---

### Official Review · Reviewer_9Koy · 2023-06-29

**Soundness:** 3 good
**Presentation:** 3 good
**Contribution:** 2 fair
**Rating:** 3
**Confidence:** 4

**Summary:**

The paper proposes three mix-up inspired regularizers for finetuning CLIP as a way to mitigate the visual-text feature gap in the CLIP feature space. The paper proposes some theory to backup why alignment of text feature space and image feature space might be a good idea. Results in cross-modal retrieval and zero-shot classification justify the method.

**Strengths:**

The results seem reasonable at first glance, although they are significantly below state-of-the-art.

The method has some novelty.

**Weaknesses:**

(1) The modality gap is a well-known phenomenon [Liang22] [CyCLIP]. I think the authors misrepresent their contributions by implying that that they discovered this phenomenon by repeated stating in the introduction that "we found .. " that the image and text embeddings occupy separate subspaces in the hypersphere (without citing the prior work) . This phenomenon was studied in  [Liang22] and  [CyCLIP] .

 - Additionally, the findings of [Liang22] are somewhat misrepresented. In Table 1 of [Liang22], those authors find that regularizing the modality gap can increase or decrease the zero-shot performance on downstream tasks. In their discussion, Liang et al. state that a larger gap "may help some fairness and zero-shot learning applications". This is not discussed in the present work.

(2) Theoretical analysis is weak:

 - Theorem 4.1 only applies to $d=2$

- Proposition applies as $\tau \rightarrow 0^+$, and in this limit, the sum of CLIP loss and $m^2$-mix regularizer reduces to the negative sum of alignment and uniformity. However, the authors already claimed that vanishing small $\tau$ is a bad thing, since "We speculate this limitation is derived from (1) ... (2) vanished learnable $\tau$ (0.01) in $\mathcal{L}_{CLIP}$." Why then would you prove something about a vanishingly small $\tau$?

(3) It is unclear to me why $m^2$-mix + CLIP is not sufficient? Why do we need the additional regularization? Does  $m^2$-mix by itself lead to consistent gains in performance?

(4) Concerns about experiments:

 - Why do you use ViT-B/32 instead of ViT-B/16? ViT-B/16 is more common [BLIP][Maple] and many papers don't even present ViT-B/32 results. This makes it hard for readers to compare the proposed modality mix-up to current state-of-the-art. The results in Table 1, 2, and 5 are significantly worse than the results in [BLIP] and [Maple].

- Table 4: Why are there only results for Pets, SVHN, and CLEVR? It is hard to determine whether these results would generalize, perhaps you could run results on more datasets, e.g. the 10 datasets used in [Maple]?

- In Table 4, $m^2$-mix leads to about 0.7 % gain over FT, while a much larger 5.2 % gain comes from $m^3$-mix and tuning $\tau$. Table 5 shows that $m^2$-mix is worse than ZS by 3 %. This severely undermines the claims in this paper, which is that cross-modality mixup (i.e. $m^2$-mix) leads to gains in generalization performance. Where there are gains, most gains seem to come from the intra-modal mix-up regularization.

[Liang22] Liang, Victor Weixin, et al. "Mind the gap: Understanding the modality gap in multi-modal contrastive representation learning." Advances in Neural Information Processing Systems 35 (2022): 17612-17625.

[CyCLIP]  Goel, Shashank, et al. "Cyclip: Cyclic contrastive language-image pretraining." Advances in Neural Information Processing Systems 35 (2022): 6704-6719.

[BLIP] Li, Junnan, et al. "Blip: Bootstrapping language-image pre-training for unified vision-language understanding and generation." International Conference on Machine Learning. PMLR, 2022.

[Maple]  Khattak, Muhammad Uzair, et al. "Maple: Multi-modal prompt learning." Proceedings of the IEEE/CVF Conference on Computer Vision and Pattern Recognition. 2023.

Minor:

In my opinion, Figure 2 is misleading. This figure displays the text and image embeddings on opposite sides of a sphere. In reality, the modality gap is not so large that text and image embeddings corresponding to the same image-caption pair fall on opposite sides of the hypersphere. So I don't think Figure 2 is a good illustration of what's really going on.

in theorem 4.1: Why is $\tilde{x} = x_1 + x_2$? shouldn't $\tilde{x}$ by the geodesic mean of the two so that $\tilde{x}$ still lies on the sphere?

Since this is a paper about mixup, the reader might expect FT+mixup to be a reasonable baseline (not feature mixup, but mixup in image input space).

**Questions:**

N/A

---

> ### Author Rebuttal · Authors · 2023-08-10
>
> > **Q1)** (1) Modality gap is a well-known phenomenon [Liang22] [CyCLIP]. The authors misrepresent their contributions by implying that they discovered this. Additionally, in Tab1 of [Liang22], authors find that regularizing gap can increase or decrease the zero-shot performances and fairness. This is not discussed in the present work.
>
> **A1)** We'd like to thank you for your valuable comments. We already noted the papers and cited them in our paper.  As you pointed out, it is also investigated by [Liang22] and others that pre-trained VLMs have separated embeddings. However, we put more of our weight on quantitative uniformity and alignment perspective to analyze the VLM’s embedding space, and we propose a principle solution, $m^2$-mix. Instead, [Liang22] mainly analyzes it via pair-wise cosine similarity and theoretical evidence. While [Goel22] conducted the experiments on uniformity and alignment, they only provided the results on the zero-shot embeddings of pre-trained CLIP and lack the fine-tuned CLIP results which are observed by us. We'll be sure to detail these differences more directly in our paper, and relax our claim about the contribution.
>
> Based on your feedback, we have designed the experiment for a fairness application, and we will attach it to the final version of our work. In terms of zero-shot learning, we confirmed that reducing the modality gap by our $m^2$/$m^3$-Mix has advantages over some cases: zero-shot transfer under distribution shift in Sec 5.4, SIMAT in Sec 5.6, and retrieval in Tab. 3 of global response. In addition, recent work [Ouali23] shows that reducing gap helps to improve base-to-new generalization and zero-shot transfer under distribution shift. These results are somewhat at odds with the finding of [Liang22]. We believe that "relationship between modality gap and downstream performance" is a controversial topic and deserves further research. However, through our experiments, we conjecture that aligning embeddings and simultaneously increasing uniformity is beneficial for a variety of downstream tasks. Therefore, we argue that fine-tuning of VL models needs to consider both alignment and uniformity, and our methods $m^{2}$-Mix achieves high alignment and uniformity with downstream task performance improvements.
>
> > **Q2)** 2-1) Thm 4.1 only applies to $d=2$, 2-2) Prop 4.2 the authors already claimed that vanishing small $tau$ is a bad thing, why then would you prove something about a vanishingly small $\tau$?
>
> 4.1. The convolution of two independent von Mises distribution is also von Mises distribution approximately [Marković12], [Mardia00]. But, it has been known that the convolution of two independent von Mises-Fisher distribution does not have an exact form. Therefore, our theoretical analysis focuses on the von Mises distribution settings. We'll include the empirical evidence for von Mises-Fisher distribution as well as von Mises distribution in our final paper
>
> 4.2. To conduct theoretical analysis on m2mix based contrastive loss, we assumed the vanishing $\tau$. This is from the consideration of the reality situation because the vanishing $\tau$ is actually the case during fine-tuning of CLIP. Specifically, the small $\tau$ (0.01) of the pre-train CLIP remains small throughout the entire fine-tuning process, unless we manually increase it with a fixed constant value. Through Prop 4.2, we wanted to claim that robust representation is learned by improving alignment and uniformity with m2mix, even in this tau-vanishing situation.
>
> > **Q3)** (3) It is unclear to me why $m^2$-mix+CLIP is not sufficient? Does $m^2$-mix by itself lead to consistent gains in performance?
>
> While the $m^2$-Mix+CLIP induces performance improvements over baseline methods in few-shot adaptation (Tab. 4) and multi-modal classification (Tab.7) in draft, thanks to your suggestion, we felt that more extensive experiments are required to validate the  $m^2$-Mix. To this end, we conducted all the additional experiments with $m^2$-Mix, not with $m^3$-Mix. On few-shot classification (Tab.1), image captioning (Tab.2), and retrieval (Tab.3) with 3 different models, our $m^2$-Mix shows consistent improvement over baselines. These results augment the empirical evidence that supports the effectiveness of $m^2$-Mix.
>
> > **Q4)** concern about experiments:
> * ViT-B/32: Because our primary goal of this study is aiming at devising a robust fine-tuning method, we conducted the experiments based on ViT-B/32 in our manuscript for fast validation of the proposed method. However, we agree with you that validations in a standardized setting (ViT-B/16) and applicability to SOTA vision-language models are needed. To address this, we accomplished additional experiments on the global response with ViT-B/16 (Tab. 1) and CoCa-ViT-L/14 [Yu22] (Tab. 2 and 3). In Tab. 1, 2, and 3, our m2-mix consistently outperforms compare to the baselines with regard to few-shot classification, captioning, and retrieval. These results verify that our method can also give favorable advantages to fine-tuning SOTA as well as a standard model backbone
> * Datasets: Thanks for pointing out where we need to improve. In Tab1. of global response, we additionally provide the results on four more transfer learning benchmark datasets and demonstrate the effectiveness of m2-mix on these datasets. We will cover all remaining datasets in the final version.
> * Effect of m2-mix: The amount of improvement depends on the datasets and tasks. m2-mix achieved 1.2% and 1.6% improvements over FT on SVHN and ImageNet in Sec 5.4, and in Tab. 1 of the general response, m2-mix outperforms FT by 8.5%, 1.8%, 4.7%, and 5.2% on CLIP-Air, CLIP-Cars, CyCLIP-Air, and CyCLIP-Cars. Importantly, our m2-mix (without intra-modal mixups) shows consistent improvements on diverse setups.
>
> *due to space limit, we'll address your minor remarks and attach reference via official comments from tomorrow.*

---

> > ### Author Response · Authors · 2023-08-10
> > **response to minor concerns and reference**
> >
> >
> > > **minor 1**) In my opinion, Figure 2 is misleading. This figure displays the text and image embeddings on opposite sides of a sphere. In reality, the modality gap is not so large that text and image embeddings corresponding to the same image-caption pair fall on opposite sides of the hypersphere. So I don't think Figure 2 is a good illustration of what's really going on.
> >
> > **A1)** Figure 2 is the real analysis of CLIP embedding based on DOSNES [Lu et al. 2019].  We adopted DOSNES (a variant of t-SNE for spherical embeddings) to naturally visualize CLIP embeddings rather than the aforementioned ones because CLIP is worked on hyperspherical embedding space. As you mentioned, the actual similarities between multi-modal embedding pairs are not negative, so the real embeddings in the hypersphere could have different patterns from our DOSNES.
> >
> > However, it is impossible to directly visualize the actual high-dimensional embedding space, and the visualization of the embedding space, which we commonly represent through dimensionality reduction algorithms such as PCA, tSNE, and UMAP, all shows a space that is significantly transformed from the actual embedding space, so we think this is the same as the DOSNES result in Fig. 2. Nevertheless, we understand your concerns and are willing to consider re-visualizing to UMAP or tSNE based on your feedback.
> >
> > > **minor 2**) in theorem 4.1: Why is $\tilde{x}=x_{1}+x_{2}$? shouldn't $\tilde{x}$ by the geodesic mean of the two so that $\tilde{x}$ still lies on the sphere?
> >
> > **A2)** Convolution of two independent von Mises distributions is approximately von Mises distribution. Therefore, the summation of x1 and x2 still lie on the hypersphere approximately, in terms of two dimensions. Therefore, our analysis on x1+x2 is still valid in the von Mises distribution case. We will add additional theoretical or empirical results for the high-dimensional case and geodesic mean case in the final version.
> >
> > > **minor 3**) Since this is a paper about mixup, the reader might expect FT+mixup to be a reasonable baseline (not feature mixup, but mixup in image input space).
> >
> > **A3)** FT + image mixup: To reflect your constructive feedback, we evaluated two representative image-level mixup strategies (Mixup [Zhang et al. 2018] and CutMix [Yun et al. 2019]) with FT and added them as baselines in Tab. 1 of the global response. Results show that $m^2$-Mix consistently outperforms those baselines. We are planning to extend them to other experiments for our final paper.
> >
> > ---
> >
> > ## Reference
> >
> > [Liang22] Liang, Victor Weixin, et al. "Mind the gap: Understanding the modality gap in multi-modal contrastive representation learning." Advances in Neural Information Processing Systems 35 (2022): 17612-17625.
> >
> > [Goel22] Goel, Shashank, et al. "Cyclip: Cyclic contrastive language-image pretraining." Advances in Neural Information Processing Systems 35 (2022): 6704-6719.
> >
> > [Ouali23] Couairon, Guillaume, et al. "Black Box Few-Shot Adaptation for Vision-Language models." Proceedings of the IEEE/CVF International Conference on Computer Vision. 2022.
> >
> > [Marković12] Marković, Ivan, and Ivan Petrović. "Bearing-only tracking with a mixture of von Mises distributions." 2012 IEEE/RSJ International Conference on Intelligent Robots and Systems. IEEE, 2012.
> >
> > [Mardia00] Mardia, Kanti V., Peter E. Jupp, and K. V. Mardia. Directional statistics. Vol. 2. New York: Wiley, 2000.
> >
> > [Yu22] Jiahui Yu, et al. "CoCa: Contrastive Captioners are Image-Text Foundation Models". Transactions on Machine Learning Research. (2022).
> > [Mardia et al. 2000] Mardia, Kanti V., Peter E. Jupp, and K. V. Mardia. Directional statistics. Vol. 2. New York: Wiley, 2000.
> >
> > [Zhang et al. 2018] Zhang, Hongyi, et al. "mixup: Beyond Empirical Risk Minimization." International Conference on Learning Representations. 2018.
> >
> > [Yun et al. 2019] Yun, Sangdoo, et al. "Cutmix: Regularization strategy to train strong classifiers with localizable features." Proceedings of the IEEE/CVF international conference on computer vision. 2019.
> >
> > [Lu et al. 2019] Lu, Yao, Jukka Corander, and Zhirong Yang. "Doubly stochastic neighbor embedding on spheres." Pattern Recognition Letters 128 (2019): 100-106.

---

> ### Comment · Reviewer_9Koy · 2023-08-15
> **rebuttal response**
>
> I thank the authors for the detailed rebuttal. After carefully reading through the rebuttal, I still do not think this manuscript is ready for publication at Neurips and may benefit from another round of review. To summarize, the following weaknesses remain unaddressed:
>
> (1) *Overclaim in abstract and intro*: (I note that ECSD agrees with this). The wording in the abstract and intro attributes the discovery of the modality gap phenomenon to the authors of the current work, when this is clearly a known phenomenon. Further, the authors state as their first contribution: "(1) We found that CLIP has a bipartite embedding structure " , when this structure has clearly already been explored in prior work such as [Liang22] (I am not an author of any such prior work). In my opinion, this is a severe exaggeration of the authors' contributions and warrants rejection.
>
> I understand that relevant work is cited, and there is some difference between the current work and prior work. But that doesn't give the authors freedom to claim credit for prior discoveries.
>
> (2) *Weak theory*: Theorem one applies only to two dimensions (a circle). Theorem 2 assumes vanishing tau, when the authors even show in their own empirical results that increasing tau from 0.01 to 0.05 is beneficial to few-shot accuracy. Aside from the unrealistic assumptions, the theorems only show that the proposed method maximizes alignment and uniformity. The theory does not say anything about improving few-shot generalization. The authors point to [Ouali23] in their rebuttal, but this is an empirical work. To the best of my knowledge, there is no theory that relates reducing modality gap to few-shot generalization. As a thought experiment, finetuning with cross-entropy will maximize alignment and uniformity and reduce the modality gap to zero, because the prototypes will become evenly distributed over the sphere and the training samples will become clustered their respective prototypes. However, in few-shot setting, it is often desirable to stop early (before alignment and uniformity are maximized) to prevent overfitting.
>
> The proposed theorems do not provide insight into how the proposed method generalizes better.
>
> (3) *Experiments*: The experiments included in the rebuttal presents drastically different results from the original manuscript. Incorporating these into the manuscript would require a some re-writing of the discussion. The authors are using a non-standard evaluation setting, and they re-run all baselines. It is not possible to compare the numbers reported by the authors with prior work. This may be fine, but it's impossible to judge how well the baseline is tuned (e.g. for few-shot finetuning using cross-entropy, it's fairly standard to tune the temperature, learning rate, batch size, how long model is trained, and how many layers are frozen/unfrozen).

---

> > ### Author Response · Authors · 2023-08-16
> > **response to response of 9Koy [1/2]**
> >
> > Firstly, let us express our appreciation to you. Through the rebuttal and discussion, we deliberate again on the position and implication of our work. We want to reply to the remaining weaknesses you raised as follows:
> >
> > * Overclaim in abstract and intro
> >   * We’d like to remark that ours and [Liang et al. 2022] are concurrent works in terms of arXiv preprint, which is why we expressed our findings that way in the paper. We'll revise the expression in the final version.
> >   * But as you already understood, our finding has differences from that of [Liang et al. 2022]: 1) quantitative analysis on uniformity and alignment; 2) such analysis on fine-tuned CLIP as well as pre-trained CLIP. Therefore, there is a new and crucial message that is lacking in other works: “After standard fine-tuning on a downstream dataset, CLIP still has a separated embedding space with poor uniformity and alignment”, which motivates us to devise the multi-modal mixup for better fine-tuning. To avoid misunderstanding, we’ll revise the expressions in a way that emphasizes the differences from other studies and puts more weight on our findings.
> >
> > * Weak theory
> >   * _Theorem4.1_) We first wanted to verify our method theoretically in a simple setup, and then empirically demonstrate it further in complex setups. While Theorem 4.1 presents the theoretical guarantee in a two-dimensional setting, thorough numerical analyses provided in supplementary material support the validity of our method in high-dimensional settings.
> >   * _Proposition 4.2_) As we explained in the rebuttal, we assume the vanishingly small $\tau$ that reflects the case of standard CLIP fine-tuning without manual $\tau$ engineering. While the manually increased $\tau$ helped generalization in some of our experiments, we analyzed the theoretical behavior in the default setting so that it could have broad implications for other works that do not conduct manual $\tau$ tuning. In the experiments of rebuttal and Sec 5.4 of the manuscript, $m^2$-Mix without $\tau$ engineering already shows consistently performs better than vanilla fine-tuning on various tasks, and Proposition 4.2 could justify such empirical successes.
> >   * _Theoretical analysis on generalization_) It is worth noting that the first work [Wang and Isola 2020] proposing the uniformity-alignment concept in uni-modal settings only provided theoretical results on the connection of contrastive loss and uniformity-alignment, and its relationship to generalization performance is empirically validated in that paper. Based on that work, there have been numerous follow-up works (>1000) including theoretical [Huang et al. 2023] and empirical [Pu et al. 2022] perspectives. Through Prop 4.2 in this work, we disclose the limiting behavior of multi-modal contrastive loss with regard to uniformity-alignment for the first time. Thus, we believe that Prop 4.2 has distinct value itself as a first step towards a theoretical connection of uniformity-alignment optimization in multi-modal contrastive learning, and expect that this analysis can be used as a strong bridge to further research.
> >
> > **(This is [1/2] comment. Due to space limit, the reference and responses to concern about experiments are presented in the next [2/2] comment)**

---

> > > ### Author Response · Authors · 2023-08-16
> > > **response to response of 9Koy [2/2]**
> > >
> > > (_continued response_)
> > >
> > > * experiments
> > >   * For the few-shot classification task, it was necessary to present experiments with a different dataset and model backbone than in the manuscript in the rebuttal to address your and other reviewers' concerns. We will revise the content of Section 5.4 based on the results of this rebuttal, and as you say, we will redo the experiments that need to be redone.
> > >   * We’d like to explain more about the experimental setup of CLIP-ViT-B/16 in few-shot classifications. Since one of our main focuses in this paper is to verify that robust adaptation is possible for a few training samples, we adopted a few-shot setup [Zhou et al. 2022] instead of experimenting with a full dataset setup, which is standard in existing CLIP fine-tuning papers [Worstman et al. 2022] [Kumar et al. 2022], so numerical comparison with previous studies is difficult. All methods except MaPLe were trained on 16-shot training samples for 200 (eurosat) and 3200 (rest) iterations per dataset, using AdamW optimizer (default parameter) and cosine learning rate scheduler, following the fine-tuning settings of the [Worstman et al. 2022] paper, and the sweep ranges of learning rate and weight decay were set to {1e-6, 3e-6, 5e-6, 7e-6, 1e-5} and {1e-1, 1e-2, 1e-3} respectively. In the experiments in rebuttal, we did not fix the temperature parameter. Except for MaPLe, all methods trained the entire model without frozen layers. Our $m^2$-Mix method was also run with the same number of iterations and the same parameter sweep range.
> > >   * For MaPLe and MaPLe+$m^2$-Mix, the image encoder and text encoder were frozen and only the prompt learner parameter was learnt for 1600 (eurosat) and 16000 (rest) iterations. Following the settings in the paper, we adopted SGD optimizer with cosine lr scheduler and explored learning rate among {0.0005, 0.001, 0.0035, 0.005}, including the proposed learning rate of 0.0035, and the weight decay parameter was also explored {0.0, 0.1} including the proposed weight decay of 0.0.
> > >
> > > ---
> > >
> > > ## Reference
> > > [Liang et al. 2022] Liang, Victor Weixin, et al. "Mind the gap: Understanding the modality gap in multi-modal contrastive representation learning." Advances in Neural Information Processing Systems 35 (2022): 17612-17625.
> > >
> > > [Wang and Isola 2020] Wang, Tongzhou, and Phillip Isola. "Understanding contrastive representation learning through alignment and uniformity on the hypersphere." International Conference on Machine Learning. PMLR, 2020.
> > >
> > > [Huang et al. 2023] Huang, Weiran, et al. “Towards the Generalization of Contrastive Self-Supervised Learning.” International Conference on Learning Representations. 2023.
> > >
> > > [Zhou et al. 2022] Zhou, Kaiyang, et al. "Learning to prompt for vision-language models." International Journal of Computer Vision 130.9 (2022): 2337-2348.
> > >
> > > [Worstman et al. 2022] Wortsman, Mitchell, et al. "Robust fine-tuning of zero-shot models." Proceedings of the IEEE/CVF Conference on Computer Vision and Pattern Recognition. 2022.
> > >
> > > [Kumar et al. 2022] Kumar, Ananya, et al. "Fine-Tuning can Distort Pretrained Features and Underperform Out-of-Distribution." International Conference on Learning Representations. 2022.
> > >
> > > [Ouali et al. 2023] Couairon, Guillaume, et al. "Black Box Few-Shot Adaptation for Vision-Language models." Proceedings of the IEEE/CVF International Conference on Computer Vision. 2023.
> > >
> > > [Pu et al. 2022] Pu, Shi, Kaili Zhao, and Mao Zheng. "Alignment-uniformity aware representation learning for zero-shot video classification." Proceedings of the IEEE/CVF Conference on Computer Vision and Pattern Recognition. 2022.

---

> > > > ### Author Response · Authors · 2023-08-20
> > > > **reviewer 9Koy**
> > > >
> > > > Dear Reviewer 9Koy,
> > > >
> > > > Could you please take a look at our rebuttal and let us know if the additional experiments and discussions address your concerns? We really appreciate your time and effort in reviewing our paper.
> > > > Besides, we are open to conducting more experiments and discussions if the reviewer provides any further suggestions.
> > > >
> > > > Yours sincerely,
> > > > Authors of paper 3682

---

### Official Review · Reviewer_ECSD · 2023-07-06

**Soundness:** 3 good
**Presentation:** 2 fair
**Contribution:** 2 fair
**Rating:** 5
**Confidence:** 4

**Summary:**


This paper studies a data augmentation method for effectively finetuning multi-modal model (i.e., CLIP). To address poor alignment of image and language space, the authors mix the embeddings of image and text while considering the geometry of the hypersphere. The authors give theoretical analysis as well as extensive experiments on several tasks including retrieval, calibration, few-/zero-shot classification.


**Strengths:**

(1) As a data augmentation strategy for contrastive learning, I think that the proposed method has novelty to some extent, different from previous arts such as i-Mix and Un-Mix. Specifically, the proposed data augmentation is performed in the embedding spaces while considering the geometry of the hyperspheres, in contrast with i-Mix and Un-Mix which mix raw images.


(2) The experiments are diverse and extensive and can validate the effectiveness of the proposed method. In particular, the performance on cross-modal retrieval performs significantly better than the competing methods.


**Weaknesses:**

(1) The first contribution regarding poor alignment of the two modal space of CLIP seems to be overclaimed.

As far as I know, Liang et al. [24] has clearly disclosed the modality gap in multi-modal contrastive representation learning, clarifying the two modalities in CLIP (and other models) are separately clustered (restricted to narrow cones) and have poor alignment and uniformity. What is difference between the authors finding and [24]? I expect an in-depth discussion in this respect and relax the claim if appropriate.

(2) Mixup-like data augmentation in feature space has been proposed in previous works including [Ref01] and [Ref02]. I suggest authors explicitly discuss the connection and difference from them.

[Ref01] Terrance DeVries and Graham W Taylor. Dataset augmentation in feature space. arXiv preprint arXiv:1702.05538, 2017a.
[Ref02] Vikas Verma, Alex Lamb, Christopher Beckham, Amir Najafi, Ioannis Mitliagkas, Aaron Courville, David Lopez-Paz, and Yoshua Bengio. Manifold mixup: Better representations by interpolating hidden states. In ICML, 2019.


**Questions:**

Please see “Weaknesses” section.

**Limitations:**

The authors have adequately discussed the limitations of the proposed method in the supplementary material.

---

> ### Author Rebuttal · Authors · 2023-08-10
>
> We are grateful for your productive feedback on our paper and your remark on the strength of our methodology and experiments.
>
> > **Q1)** The first contribution regarding poor alignment of the two modal space of CLIP seems to be overclaimed. As far as I know, Liang et al. [24] has clearly disclosed the modality gap in multi-modal contrastive representation learning, clarifying the two modalities in CLIP (and other models) are separately clustered (restricted to narrow cones) and have poor alignment and uniformity. What is difference between the authors finding and [24]? I expect an in-depth discussion in this respect and relax the claim if appropriate.
>
> **A1)** As you pointed out, it is already well clarified by [Liang et al. 2022] that pre-trained vision-language models have separated embeddings for each modality. While their finding is mainly based on the pair-wise cosine similarity and theoretical analysis, we weigh our attention on quantitative measurements of uniformity and alignment (Fig 2. of our manuscript), which are lacking in the previous work. While the author of CyCLIP [Goel et al. 2022] analyzed the uniformity and alignment of CLIP embeddings, they only considered the zero-shot embedding of CLIP. Meanwhile, our observation includes the fine-tuned CLIP (which is lacking in the CyCLIP paper) as well as pre-trained CLIP. We will be sure to detail these differences more directly in our paper, and soften our claim about the finding on separated embeddings in pre-trained CLIP. Thanks for the great feedback!
>
> > **Q2)** Mixup-like data augmentation in feature space has been proposed in previous works including [Ref01] and [Ref02]. I suggest authors explicitly discuss the connection and difference from them. [Ref01] Terrance DeVries and Graham W Taylor. Dataset augmentation in feature space. arXiv preprint arXiv:1702.05538, 2017a. [Ref02] Vikas Verma, Alex Lamb, Christopher Beckham, Amir Najafi, Ioannis Mitliagkas, Aaron Courville, David Lopez-Paz, and Yoshua Bengio. Manifold mixup: Better representations by interpolating hidden states. In ICML, 2019.
>
> **A2)** Thank you for suggesting valuable related works. Our methods and the related works, [Ref01] [Ref02], exploit the data augmentation in feature space, not input space. However, there are two contributions to our methods. First, we propose a hidden feature mixup method on heterogeneous modality. When there are text and image feature spaces, our $m^{2}$-Mix explores inter-relationship by mixing image and text features, while the previous methods explore intra-relationship by mixing image features themselves. Second, our methods interpolate between data points on the hypersphere, not Euclidean space. As shown in Equation (4) of the main paper, we interpolate the data points by geodesic. As a result, the augmented features by our methods lie on the hypersphere. Self-supervised methods project the embedding on the hypersphere; therefore, it is important to guarantee that the augmented features are on the hypersphere. Table 6 indicates the importance of geodesic interpolation.
>
> In other words, our method is a more generalized Mixup that handles the multimodal and hypersphere spaces. From Equation (4) of the main paper, our method degenerates to the traditional Mixup or manifold Mixup if we change a and b as the same domain (text or image) and its coefficient as just $\lambda$ instead of the sinusoidal function with $\lambda$ and $\theta$.
>
> ---
>
> [Liang et al. 2022] Liang, Victor Weixin, et al. "Mind the gap: Understanding the modality gap in multi-modal contrastive representation learning." Advances in Neural Information Processing Systems 35 (2022): 17612-17625.
>
> [Goel et al. 2022] Goel, Shashank, et al. "Cyclip: Cyclic contrastive language-image pretraining." Advances in Neural Information Processing Systems 35 (2022): 6704-6719.

---

> > ### Comment · Reviewer_ECSD · 2023-08-18
> >
> > I thank the authors for their detailed responses. The authors alleviated but not fully addressed my concern about the first contribution; I also notice that Reviewer 9Koy has the same concern. As such, I keep my original rating  (5) unchanged.

---

### Official Review · Reviewer_yWq1 · 2023-07-06

**Soundness:** 3 good
**Presentation:** 3 good
**Contribution:** 3 good
**Rating:** 5
**Confidence:** 4

**Summary:**

This paper proposes a new method for robust fine-tuning called Geodesic Multi-Modal Mixup. The method improves the uniformity and alignment in multi-modal learning, thereby enhancing the performance of downstream tasks. Previous research has shown promising performance of large-scale pre-trained models on various downstream tasks, but the analysis of embedded representations for multi-modal tasks is still insufficient, and the transferability of cross-modal tasks needs to be improved. The motivation of this paper is to address these limitations by introducing the Geodesic Multi-Modal Mixup method, which demonstrates improved performance on various downstream tasks through experiments.

**Strengths:**

1. The paper provides a new perspective for understanding multi-modal embeddings in terms of uniformity and alignment.

2. The paper proposes a new end-to-end fine-tuning method for improving the robustness and transferability of multi-modal representations for downstream tasks.

3. The proposed method, Geodesic Multi-Modal Mixup, is shown to learn a more robust representation and provide improved performance on various downstream tasks such as retrieval, classification, and structure-awareness.

**Weaknesses:**

1. The paper highlights that the analysis of learned embeddings and transferability for cross-modal tasks has not been explored well, i.e., image captioning, visual questions answering tasks, and so on.

2. The multiple multi-modal mixup seems too complex, would it cast shadow on the learning and generalization ability? More multi-modal backbones like the variants of CLIP or other vision-language models like DeCLIP, FILIP, CLOOB, CyCLIP. Since the approach is agnostic to the architecture and models, I would encourage the authors to perform additional experiments to demonstrate the effectiveness of the proposed approach on other vision-language models.


**Questions:**

see weakness above

**Limitations:**

see weakness above

---

> ### Author Rebuttal · Authors · 2023-08-10
>
> Thank you so much for your valuable feedback and that you find value in our work. We agree with your concerns and have conducted additional experiments based on your feedback.
>
> ---
> > **Q1)** The paper highlights that the analysis of learned embeddings and transferability for cross-modal tasks has not been explored well, i.e., image captioning, visual questions answering tasks, and so on.
> > **Q2)** The multiple multi-modal mixup seems too complex, would it cast shadow on the learning and generalization ability? More multi-modal backbones like the variants of CLIP or other vision-language models like DeCLIP, FILIP, CLOOB, CyCLIP. Since the approach is agnostic to the architecture and models, I would encourage the authors to perform additional experiments to demonstrate the effectiveness of the proposed approach on other vision-language models.
>
> **A2)**
> Despite its effectiveness, as you noticed, $m^3$-Mix has multiple components contributing to learning and generalization, so it seems hard to clear out the core factor of learning and generalization. To clearly determine the effect of our main contribution ‘multi-modal mixup’, we perform all additional experiments in this rebuttal with $m^2$-Mix.
>
> **A1, 2)**
> In our manuscript, we performed the following tasks: (uni-modal) few-shot image classification under general and distribution shift settings; (multi-modal) retrieval, sentiment classification, and embedding arithmetic. To further validate our method, we evaluate our method on the image captioning task with CoCa [Yu et al. 2022] model. We fine-tune CoCa on MS COCO dataset via CoCa objective (contrastive and captioning loss), and showcase the results of image captioning and retrieval in Table 2. and Table 3. of the global response, respectively. Our $m^2$-Mix consistently outperforms the standard fine-tuning method with regard to almost all metrics of interest. From this, we confirm that our $m^2$-Mix not only benefits the discriminative tasks but also the generative task.
>
> **A2)**
> Besides, in Table 1., we evaluate our method on CyCLIP [Goel et al. 2022] for few-shot image classification. Here, $m^2$-Mix also successfully improves performance across all considered datasets. This further demonstrates that our $m^2$-Mix is a versatile fine-tuning approach that can be seamlessly integrated across VL models (CLIP, CyCLIP, CoCa, and so on).
>
> ---
>
> [Yu et al. 2022] Jiahui Yu, et al. "CoCa: Contrastive Captioners are Image-Text Foundation Models". Transactions on Machine Learning Research. (2022).
>
> [Goel et al. 2022] Goel, Shashank, et al. "Cyclip: Cyclic contrastive language-image pretraining." Advances in Neural Information Processing Systems 35 (2022): 6704-6719.

---

> ### Author Response · Authors · 2023-08-20
> **reviewer yWq1**
>
> Dear Reviewer yWq1,
>
> Could you please take a look at our rebuttal and let us know if the additional experiments and discussions address your concerns? We really appreciate your time and effort in reviewing our paper.
> Besides, we are open to conducting more experiments and discussions if the reviewer provides any further suggestions.
>
> Yours sincerely,
> Authors of paper 3682

---

### Official Review · Reviewer_uSAw · 2023-07-06

**Soundness:** 3 good
**Presentation:** 3 good
**Contribution:** 3 good
**Rating:** 4
**Confidence:** 4

**Summary:**

This paper addresses the problem of improving vision-language representation learning using feature-space augmentation. The authors claim that approaches such as CLIP have led to poor alignment between text features and image features, and the space between them lacks uniformity. They propose an approach called Geodesic multimodal mix-up to mix vision and language embeddings in order to generate hard negative samples. These hard negative samples, along with the original negative and positive samples, are then used in existing contrastive algorithms. The authors provide a theoretical guarantee of the difficulty of these examples. They evaluate their approach on several common benchmarks such as MS-COCO/Flickr30K (retrieval) and Pets/SVHN/CLEVR (classification). Compared to standard mix-up techniques such as linear combination and re-normalization, their approach consistently shows improvements across different settings.





**Strengths:**


The paper is clearly written and easy to follow. The approach is simple but intuitively makes sense, i.e., mixing up embeddings while still residing on the normalized surface should be better than linear combination and re-normalization. The authors also provided a sound theoretical justification for their approach. The gains on the chosen benchmark over the previous approaches are decent.

**Weaknesses:**

The improvement was shown in vanilla settings, for example, with basic CLIP and small datasets. It is not clear if such a mix-up scheme holds up in other large-scale settings where the data distribution might be different. Also, reported numbers are far from SOTA.

**Questions:**

N/A

---

> ### Author Rebuttal · Authors · 2023-08-10
>
> We appreciate you for your attention and valuable comment. We agree with your concerns and have conducted additional experiments to address them.
>
> ---
> > **Q)** The improvement was shown in vanilla settings, for example, with basic CLIP and small datasets. It is not clear if such a mix-up scheme holds up in other large-scale settings where the data distribution might be different. Also, reported numbers are far from SOTA.
>
> **A)**
>
> In this work, we focus on devising a robust fine-tuning method for pre-trained VL models. Here, we believe that the method (as a ‘robust’ fine-tuning approach) should have the following desirable properties: 1. Achieving stable performance on the data-scarce regime (Sec 5.4); 2. Robustness under distribution shift (Sec 5.4); 3. Robustness under modality missing (Sec 5.5); 4. calibration (Sec 5.2), and 5) preserving semantic embedding structures  (Sec 5.6). Therefore, we spared much of our attention to these components rather than the scale of experiments or SOTA performance, and we demonstrated that our method achieve all the above properties from each section.
>
> Nonetheless, as you pointed out, it is crucial to determine whether our method is effective for large-scale SOTA VL backbones beyond the basic CLIP. To this end, we evaluate our method on the one of SOTA VL models, CoCa [Yu et al. 2022], with CoCa-Large configuration governing 787M parameters which are 9 times larger than the CLIP-ViT-Base setting in our manuscript. We fine-tuned the pre-trained CoCa model on the MS COCO dataset via CoCa’s pre-train objective with $m^2$-Mix-based contrastive loss. In Table 2. and Table 3. of the global response, our $m^2$-Mix consistently outperforms compare to the original CoCa fine-tuning baselines with regard to captioning evaluation metrics and recalls of retrieval. These results verify that our method can give favorable advantages on large-model fine-tuning in terms of both generative and discriminative tasks.
>
> Moreover, we compare our method with SOTA parameter-efficient fine-tuning approaches such as MaPLe [Khattak et al. 2023] both in our manuscript (Sec 5.4) and in this rebuttal (Table 1.). In these experiments, our method constantly improves performance, so it demonstrates its effectiveness and versatility as a plug-and-play module.
>
> ---
>
> [Yu et al. 2022] Jiahui Yu, et al. "CoCa: Contrastive Captioners are Image-Text Foundation Models". Transactions on Machine Learning Research. (2022).
>
> [Khattak et al. 2023] Khattak, Muhammad Uzair, et al. "Maple: Multi-modal prompt learning." Proceedings of the IEEE/CVF Conference on Computer Vision and Pattern Recognition. 2023.

---

> ### Author Response · Authors · 2023-08-20
> **reviewer uSAw**
>
> Dear Reviewer uSAw,
>
> Could you please take a look at our rebuttal and let us know if the additional experiments and discussions address your concerns? We really appreciate your time and effort in reviewing our paper.
> Besides, we are open to conducting more experiments and discussions if the reviewer provides any further suggestions.
>
> Yours sincerely,
> Authors of paper 3682

---

### Author Rebuttal · Authors · 2023-08-10

# Global Response
We thank the reviewers for taking the time to review our paper and for your valuable feedback. We have carefully considered your comments and reflected in our response. In this global response, 1) we first provide a brief review of our draft and then 2) provide a summary of additional statements that we’ve made to address your feedback and suggestions during the rebuttal period. Finally, 3) we elaborate on settings of additional experiments(**Tables and Figures are attached to pdf**).
- - -
## 1. Review of research highlights
* We observed that both the pre-trained CLIP and its naively fine-tuned counterpart  have separated embedding space for each modality and show quantitatively poor alignment and uniformity scores, which may limit the transferability of learned embedding.
* From our findings, we devised a new fundamental approach ‘geodesic multi-modal mixup ($m^2$-Mix)’-based contrastive learning for robust fine-tuning to enhance the alignment and uniformity of embeddings.
* We provide two theoretical analyses of the proposed method to support its validity
  * hardness-guarantee of the generated sample by $m^2$-Mix, which is crucial for the success of contrastive learning.
  * asymptotic behavior of standard CLIP loss with $m^2$-Mix that maximizes the alignment and uniformity with few assumptions.
* Through extensive experiments on retrieval, calibration, few-shot adaptation, zero-shot transfer under distribution shift, multi-modal classification under modality missing, and embedding arithmetic, we demonstrate that the proposed method effectively improves performance over baseline methods.

## 2. Additional remarks from the rebuttal period
During the rebuttal period, we conducted several additional experiments to address the reviewers' concerns and suggestions. Here, we provide our new findings from the results of extra experiments in summary:
* (9Koy) Without the aid of intra-modal mixup regularization such as V/L/VL-Mix, our $m^2$-Mix independently yields consistent performance gains on a variety of tasks, and in some cases the gains are significant (+5~8% Acc).
* (9Koy) We evaluate our $m^2$-Mix with ViT-B/16 (not B/32 of draft) on four additional transfer learning benchmarks, and $m^2$-Mix achieves consistent performance improvements.
* (yWq1) Beyond the basic CLIP, we demonstrated that our $m^2$-Mix is helpful to other recent vision-language models (VLM) such as CyCLIP [Goel et al. 2022] and CoCa [Yu et al. 2022].
* (yWq1) Our method gives an advantage not only for retrieval, multi-modal classification, and embedding arithmetic, but also for generative tasks such as image captioning.
* (uSAw, 9Koy) $m^2$-Mix brings benefits to large-scale state-of-the-art VLM (e.g., CoCa-Large that has 9 times more parameters compared to CLIP-Base) as well as standard CLIP.
* (iPuA) We tweaked the parameters of the Beta distribution and found that our $m^2$-Mix performs optimally between 0.1 and 0.5, just like the standard Mixup. This suggests that when choosing a mixed sample, mixing one side more than the other, rather than half-and-half, leads to more stable learning, which contributes to better generalization performance.

## 3. Experimental setup
We evaluated our method on three tasks (few-shot classification, image captioning, and retrieval), and the experimental setup for each task is as follows:

### 3.1. few-shot classification
We consider four standard benchmark datasets (EuroSAT, FGVC Aircraft, UCF101, and Stanford Cars) and two VLMs with different backbone and pre-trained weights (CLIP-ViT-B/32 and CyCLIP-ResNet50 with official pre-trained checkpoints) that are not included in our manuscript.
We adopt the same setting with Sec 5.4 of our draft to implement the contrastive learning-based fine-tuning on these image-label paired datasets, i.e., promptizing a label with a common context “a photo of {classname}” and regard this as a caption per images. As baselines, we consider zero-shot inference, vanilla fine-tuning with traditional image space mixups (Mixup [Zhang et al. 2018] and Cutmix [Yun et al. 2019]), robust fine-tuning method (WiSE-FT [Wortsman et al. 2022]), and a state-of-the-art (SOTA) parameter-efficient fine-tuning method (MaPLe [Khattak et al. 2023]). All experiments are done under a 16-shot training set and full test set, and top-1 accuracy is reported.
###  3.2. image captioning and retrieval
We adopt CoCa-ViT-L/14 with pre-trained checkpoint `laion2b_s13b_b90k` from the OpenCLIP library as our VLM to demonstrate the effectiveness of our method on a large-scale SOTA model. We fine-tune CoCa on MS COCO training set with 1) only captioning loss (this is a strategy that CoCa’s authors adopted), 2) contrastive loss + captioning loss, and 3) contrastive loss + $m^2$-Mix + captioning loss. After training, we evaluate the models in terms of captioning metrics such as BLEU4, CIDEr, etc  (on COCO) and retrieval recalls  (on COCO and Flickr30k).
- - -
[Goel et al. 2022] Goel, Shashank, et al. "Cyclip: Cyclic contrastive language-image pretraining." Advances in Neural Information Processing Systems 35 (2022)

[Yu et al. 2022] Jiahui Yu, et al. "CoCa: Contrastive Captioners are Image-Text Foundation Models". Transactions on Machine Learning Research. (2022).

[Zhang et al. 2018] Zhang, Hongyi, et al. "mixup: Beyond Empirical Risk Minimization." International Conference on Learning Representations. 2018.

[Yun et al. 2019] Yun, Sangdoo, et al. "Cutmix: Regularization strategy to train strong classifiers with localizable features." Proceedings of the IEEE/CVF international conference on computer vision. 2019.

[Wortsman et al. 2022] Wortsman, Mitchell, et al. "Robust fine-tuning of zero-shot models." Proceedings of the IEEE/CVF Conference on Computer Vision and Pattern Recognition. 2022.

[Khattak et al. 2023] Khattak, Muhammad Uzair, et al. "Maple: Multi-modal prompt learning." Proceedings of the IEEE/CVF Conference on Computer Vision and Pattern Recognition. 2023.

---

### Decision · Program_Chairs · 2023-09-21

**Decision:**

Accept (poster)

**Comment:**

This submission has received five reviews with mixed ratings ranging from 3 to 7. The reviews were answered with a rebuttal and then discussed between the authors and reviewers. This paper was then discussed in depth by the area and senior are chair.

The reviewers list as positive points that the approach presents a novel perspective and contains some novelty. One reviewer asks for experiments with a larger backbone that is answered to in the rebuttal but did not find influence on the respective review score. There are other minor technical questions all of which have been answered in the rebuttal as well.

The main remaining discussion point is the claim of novelty with respect to Liang et al. 2022. We understand that this may have been concurrent work at arXiv, however that is more than one and a half year ago, so while novelty claims in the arXiv version may be justified in a conference version those passages in the text need to be adjusted. It is important to note that this concerns the text of the paper, we acknowledge that in the experiments a comparison to [24] has been included, so this is not a case where prior work has been ignored. Therefore in the discussion we concluded that the reject recommendation of reviewer 9KoY that is based mainly on this point is too strong. We believe based on the rebuttal that the authors are willing to reword carefully their discoveries with respect to Liang, [24] and their own arXiv work.

Besides this dispute most other points are answered to and the paper presents a new perspective on a relevant topic. Thus we decided to down-weight  the most negative score and recommend acceptance of the paper. The authors are expected to work the discussion and the rebuttal into the final camera ready submission.